# Integrative functional genomics analysis identifies pleiotropic genes for vascular diseases

Charles U. Solomon [1,2,3,12], David G. McVey [1,2,3,12], Catherine Andreadi[1,2,3], Peng Gong[1,2,3], Lenka Turner[1,2,3], Dedrick S. S. Song [4], Heming Zhang[4], Dominic P. Lee [4], Elisavet Karamanavi[1,2,3], Wei Yang[5,6], Jiapeng Chu[4], Runji Chen[5,7], Kim E. Haworth[1,2,3], Chukwuemeka George Anene-Nzelu[4,8], Hui Li[4], Matthew J. Denniff[1,2,3], Peter Y. Li[4], Yanhong Zhang[5], Xiaoxin Huang[5], Gavin E. Morris[1,2,3], Peter A. Greer[9], Emma J. Stringer[1,2,3], Haojie Yu [4], Roger S. Y. Foo [4], Gillian Douglas[10,11], Nilesh J. Samani[1,2,3,13], Tom R. Webb [1,2,3,13] & Shu Ye [1,2,3,4,5,7,13] ✉

Several vascular diseases including coronary artery disease, hypertension, stroke, and abdominal aortic aneurysm, have significant genetic underpinnings. Genome-wide association studies have unveiled many genetic loci associated with one or more of these diseases. However, the causative genes at most of these loci are yet to be determined, which hampers the translation of the genetic findings into a better understanding of the disease mechanisms and the identification of new therapeutic targets. Here, in an integrative functional genomics analysis of these loci, we identify a panel of likely causal genes, some of which are pleiotropic for more than one of these vascular diseases. Pooled CRISPR knockout screen analyses of these likely causal genes indicate that many of them influence vascular smooth muscle cell behaviour, and validation experiments of selected genes confirm that *FES*, *BCAR1*, *CARF* and *SMARCA4* exert such effects. Further functional experiments focusing on *FES*, a pleiotropic gene for both coronary artery disease and hypertension, show that it modulates the expression of genes involved in vascular remodeling and that *Fes* knockout in mice promotes atherosclerosis as well as raises blood pressure. These findings provide an insight into the genetic basis of vascular diseases and inform targets for therapeutic development.

In recent years, genome-wide association studies (GWAS) have unveiled genomic loci where genetic variants are associated with vascular diseases such as coronary artery disease (CAD)[1–4], hypertension[5], stroke[6,7], and abdominal aortic aneurysm (AAA)[5,8]. To facilitate the medical translation of these genetic discoveries, it is important to identify and investigate the causal genes at these loci.

Vascular smooth muscle cells (VSMC) are a major player in vascular diseases. VSMC phenotypic switching, migration, proliferation and apoptosis play important roles in the pathogenesis of atherosclerosis, which promotes CAD, hypertension, stroke, and AAA[9]. Further, VSMC hypercontractility and vascular remodelling involving VSMC migration, proliferation and apoptosis contribute to elevated blood pressure[10], VSMC loss and impaired contractility are hallmarks

---

of AAA[11], while VSMC degeneration causes cerebral small-vessel disease that can lead to ischaemic and haemorrhagic strokes[12].

As most disease-associated genetic variants revealed by GWASs are located in non-coding regions of the genome[13,14], a prevailing hypothesis is that these variants can alter the individual's disease risk through their effect on gene expression[13,14]. Studies have shown that there is considerable inter-individual variation in VSMC behaviour, largely due to genetic variation among individuals[15,16]. Interestingly, some vascular disease-associated genetic variants have been reported to modulate gene expression in VSMCs and affect VSMC behaviour[15–18]. Therefore, it is likely that the genetic influence on vascular disease acts partly through VSMCs.

Here, we report results from an integrative functional genomics analysis on VSMCs (outlined in Fig. 1), with the aim to identify causal genes at the genomic loci that have been found to be associated with CAD, hypertension, stroke and/or AAA in GWAS.

## Results

### Influences of vascular disease-associated genetic variants on gene expression

We recently established a large collection of VSMCs from different individuals ($n = 1486$), and on this cell bank, we performed genome-wide genotyping and whole-genome RNA-sequencing (RNA-seq) to generate a comprehensive catalogue of expression quantitative trait loci (eQTL) with stringent statistical significance[16]. We compiled a list of tagging single nucleotide polymorphisms (SNPs) that marked the genetic variants/haplotypes reported to be associated with CAD in GWAS[1–4] and then interrogated these tagging SNPs in the VSMC eQTL catalogue. As reported previously[16], we observed that ~60% of the CAD-associated SNPs had an eQTL effect on gene expression, with two-thirds of them affecting more than one gene.

Here we compiled lists of tagging SNPs that marked the genetic variants/haplotypes reported to be associated with blood pressure/hypertension[5,19], stroke[5–7] and/or AAA[5,8] in GWAS, and interrogated these tagging SNPs in the abovementioned VSMC eQTL catalogue. We found that ~ 59% of the blood pressure/hypertension-associated SNPs, ~ 55% of the stroke-associated SNPs, and ~ 59% of the AAA-associated SNPs, had eQTL effects in VSMCs (Supplementary Data 1-3 and Supplementary Fig. 1). Of these SNPs, ~ 66% showed eQTL effects on more than one gene. Of the associated genes, ~ 77% were protein-coding, and ~ 15% were long non-coding RNAs (lncRNAs) (highlighted in green and yellow, respectively, in Supplementary Data 1–3). When male samples ($n = 780$) and female samples ($n = 706$) were analysed separately, the eQTL effects of approximately 50% of CAD-, blood pressure/

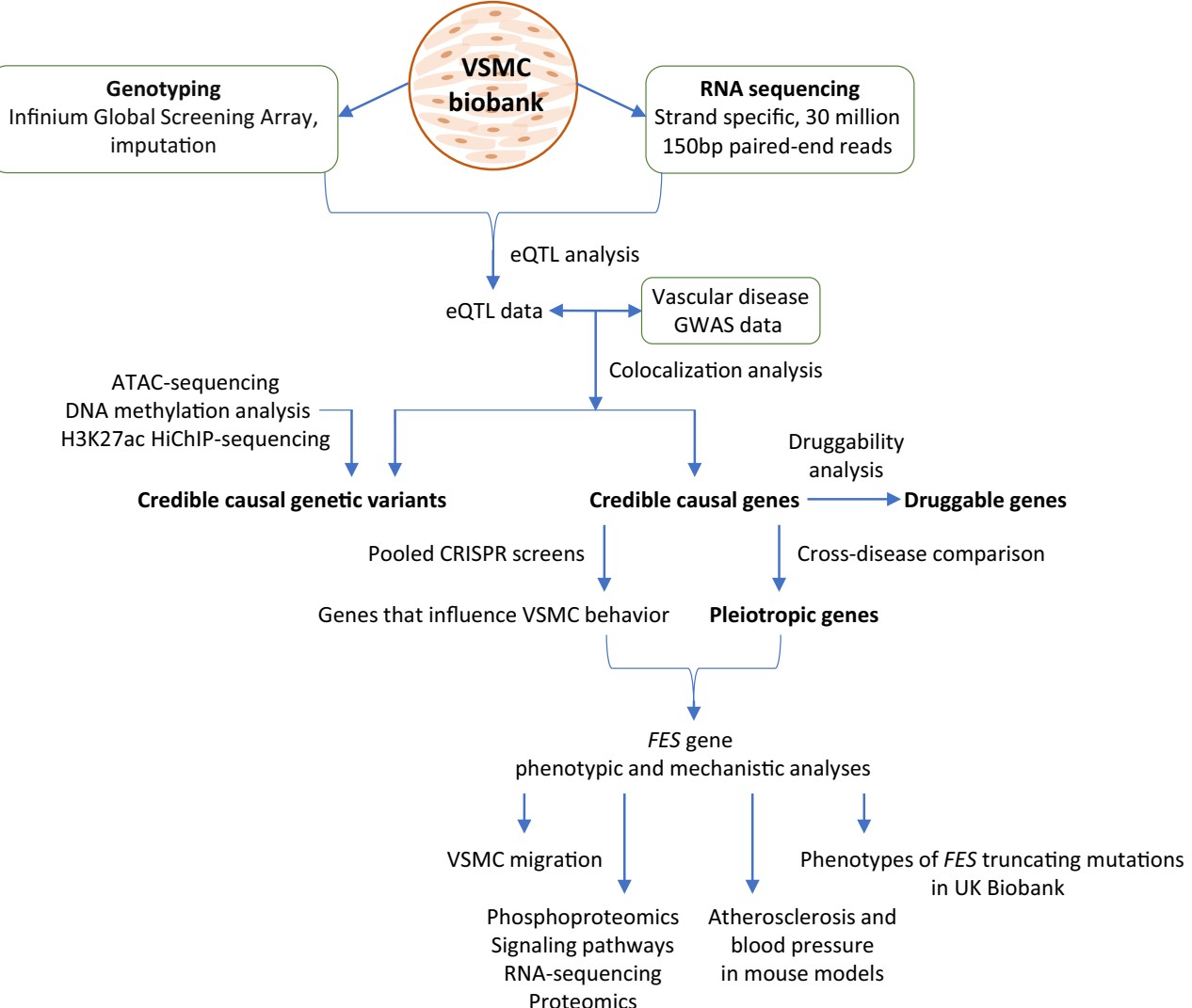

**Fig. 1 | Flow diagram of the study design.** ATAC-sequencing: assay for transposase-accessible chromatin with sequencing; eQTL: expression quantitative trait locus; GWAS: genome-wide association study; H3K27ac: histone 3 lysine 27 acetylation; HiChIP: HiC (High-throughput chromosome conformation capture) with ChIP-seq (Chromatin immunoprecipitation with sequencing); VSMC: vascular smooth muscle cells.

hypertension-, stroke-, or AAA-associated SNPs were still detected (Supplementary Data 1–4), despite the reduction in the sample sizes per group, indicating that they are common to both sexes.

### Likely causative variants/genes for CAD, hypertension, stroke, and AAA

To explore causal variants and related causal genes for each of the vascular diseases mentioned above, we performed colocalization tests, including eCAVIAR (eQTL and GWAS CAusal Variant Identification in Associated Regions)[20] and SMR/HEIDI (summary-data-based Mendelian Randomisation/Heterogeneity in Dependent Instruments)[21], utilising reported disease GWAS summary statistics in conjunction with the above-described VSMC eQTL dataset. These analyses detected panels of likely causal variants and likely causal genes for each of these vascular diseases. Supplementary Data 5–8 show the results of these colocalization analyses, including the identified likely causal variants (columns A-F), the identified likely causal genes together with eQTL data in relation to the likely causal variants (columns G-M), summary statistics of reported associations of the likely causal variants with the relevant vascular diseases (columns N-R), and the statistics of the eCAVIAR and SMR/HEIDI analyses (columns S-Y).

To investigate if any of the likely causal variants identified in the above colocalization analyses were located in transcriptionally active regions of the genome, we performed VSMC ATAC-seq (assay for transposase-accessible chromatin with sequencing) to delineate open chromatin regions, as well as DNA methylation assay and H3K27ac HiChIP-seq (histone H3 acetylation at lysine 27 high throughput chromosome conformation capture with chromatin immunoprecipitation with sequencing) since DNA demethylation and H3K27 acetylation upregulate gene expression. We interrogated the likely causal variants in regions of ATAC-seq peaks, low DNA methylation and H3K27ac, respectively. The analyses showed that a large proportion of the likely causal variants were located in regions of ATAC-seq open chromatin (Supplementary Data 9), low DNA-methylation (variants highlighted in yellow in Supplementary Data 10), and/or H3K27ac (Supplementary Data 11). The likely causal variants in relation to peaks of ATAC-seq, low DNA methylation and H3K27ac, as well as their associated eQTL genes and vascular diseases, are summarised in Supplementary Data 12. As an example, we found that the CAD-associated variant rs1894401 in the *FES* gene resided in an ATAC-seq open chromatin region with lessened DNA-methylation (Supplementary Figs. 2, 3) and with H3K27ac (Supplementary Fig. 4).

Because a genetic variant could possibly exert a long-range effect on the transcription of a gene located at a distance due to DNA looping, we utilised the VSMC H3K27ac HiChIP-seq data to investigate if any of the likely causal variants resided at a DNA interaction anchor, with a likely causal gene located at a linked DNA interaction anchor. This analysis detected multiple variant-gene pairs where a likely causal variant was detected within an anchor and a likely causal gene was found within a linked anchor (Supplementary Data 11 and Supplementary Fig. 4), suggesting that the likely causal gene was brought physically closer to the likely causal variant through DNA looping. For instance, the CAD-associated variant rs1894401 was detected in an anchor, and the *FES* gene 5′-region was observed in a linked anchor (Supplementary Fig. 4A).

The aforementioned colocalization analyses identified 134 likely causal genes for CAD, whose eQTL signals in VSMCs showed significant colocalization with CAD GWAS signals, including 84 genes that we reported previously[16] and a further 50 genes now identified using more recently available CAD GWAS data[1] (Fig. 2, Supplementary Fig. 5A and Supplementary Data 5). Further, the analyses identified 74 likely causal genes for hypertension (Fig. 2, Supplementary Fig. 5B and Supplementary Data 6), 13 likely causal genes for stroke (Fig. 2, Supplementary Fig. 5C, and Supplementary Data 7), and 43 likely causal genes for AAA (Fig. 2, Supplementary Fig. 5D and Supplementary Data 8).

In addition to the abovementioned colocalization analyses using eQTL data from the VSMC bank, we performed additional colocalization analyses using eQTL data from coronary artery and aortic artery tissues (from the GTEx Portal) together with the disease GWAS statistics. Noteworthily, most of the likely causal genes identified in the colocalization analyses using eQTL data from the VSMC bank were also significant in the colocalization analyses using eQTL data from coronary artery and aortic artery tissues (Supplementary Data 13), further indicating that they are likely to be causal genes.

Gene ontology enrichment analyses of the likely causal genes identified showed enrichment in several major biological processes and pathways, including transforming growth factor-beta (TGFβ) signalling in CAD (Supplementary Fig. 6), vascular endothelial growth factor (VEGF) signalling in hypertension (Supplementary Fig. 7), and signalling pathways of vascular endothelial growth factor receptor (VEGFR), platelet-derived growth factor receptor (PDGFR), integrin and cadherin in AAA (Supplementary Fig. 8), suggesting that the genetic influence on these vascular diseases is, at least partly, mediated by these genes acting through these biological pathways.

### Pleiotropic genes implicated in more than one vascular disease

A cross-trait linkage disequilibrium score regression analysis of GWAS summary statistics revealed significant genetic correlations between the various vascular diseases investigated in this study, indicating shared genetic influences (Fig. 3A and Supplementary Data 14).

Further, comparing the likely causal genes identified for CAD, hypertension, stroke and AAA, respectively, we noticed that 18 genes were common for two or more of these diseases, suggesting that they have pleiotropic effects. Specifically, *ARNTL*, *CSNK2B*, *FES*, *SNF8*, *TARID*, and *TCF21* were detected for both CAD and hypertension; *CARF*, *NBEAL1*, and *PROCR* were common in CAD and stroke; *BCAR1*, *ILF3-DT*, *LPAL2*, and *SMARCA4* were shared by CAD and AAA; while *PLCE1-AS1* was found for CAD and hypertension as well as AAA (Fig. 3B). In addition, *ADO* was identified for both hypertension and AAA, whilst *SURF1* and 2 lncRNA genes were detected for both hypertension and stroke (Fig. 3B).

### Deleterious protein-coding variants in likely causal genes

The above colocalization analyses focused on regulatory SNPs that influenced gene expression. Next, we further explored if there also existed deleterious variants (such as frameshift mutations and deleterious missense substitutions) in the protein-coding regions of the likely causal genes that we had identified and if these variants were associated with the diseases under this study, because such evidence would further indicate that these genes were causal. Thus, we interrogated the likely causal genes in databases from the UK Biobank[22,23] and Finnish Biobank[24] (through the AstraZeneca PheWAS Portal[25] and the PheWeb browser, respectively) for deleterious coding variants identified by exome sequencing. This interrogation identified such variants in several of the likely causal genes, including the *ADAMTS7*, *FES*, *ICA1L*, *MIA3*, and *MTAP* genes in relation to CAD and the *ADO*, *FES*, and *MAP4* genes in association with hypertension (Supplementary Data 15). However, no coding variants in any gene were associated with stroke or AAA at the genome-wide significance level ($P < 1 \times 10^{-8.7}$ in PheWAS and $P < 5 \times 10^{-8}$ in PheWeb) in the UK Biobank and Finnish Biobank databases.

### Druggability of likely causal genes

An interrogation of the likely causal genes in the Drug Gene Interaction Database (DGIdb)[26] identified 63 potentially druggable targets for CAD, 39 for hypertension, 5 for stroke, and 25 for AAA (Fig. 4 and Supplementary Data 16–23). Notably, 5 of these targets were applicable for both CAD and hypertension, 2 for both CAD and stroke, 2 for both CAD and AAA, and 1 for both hypertension and AAA (Fig. 4).

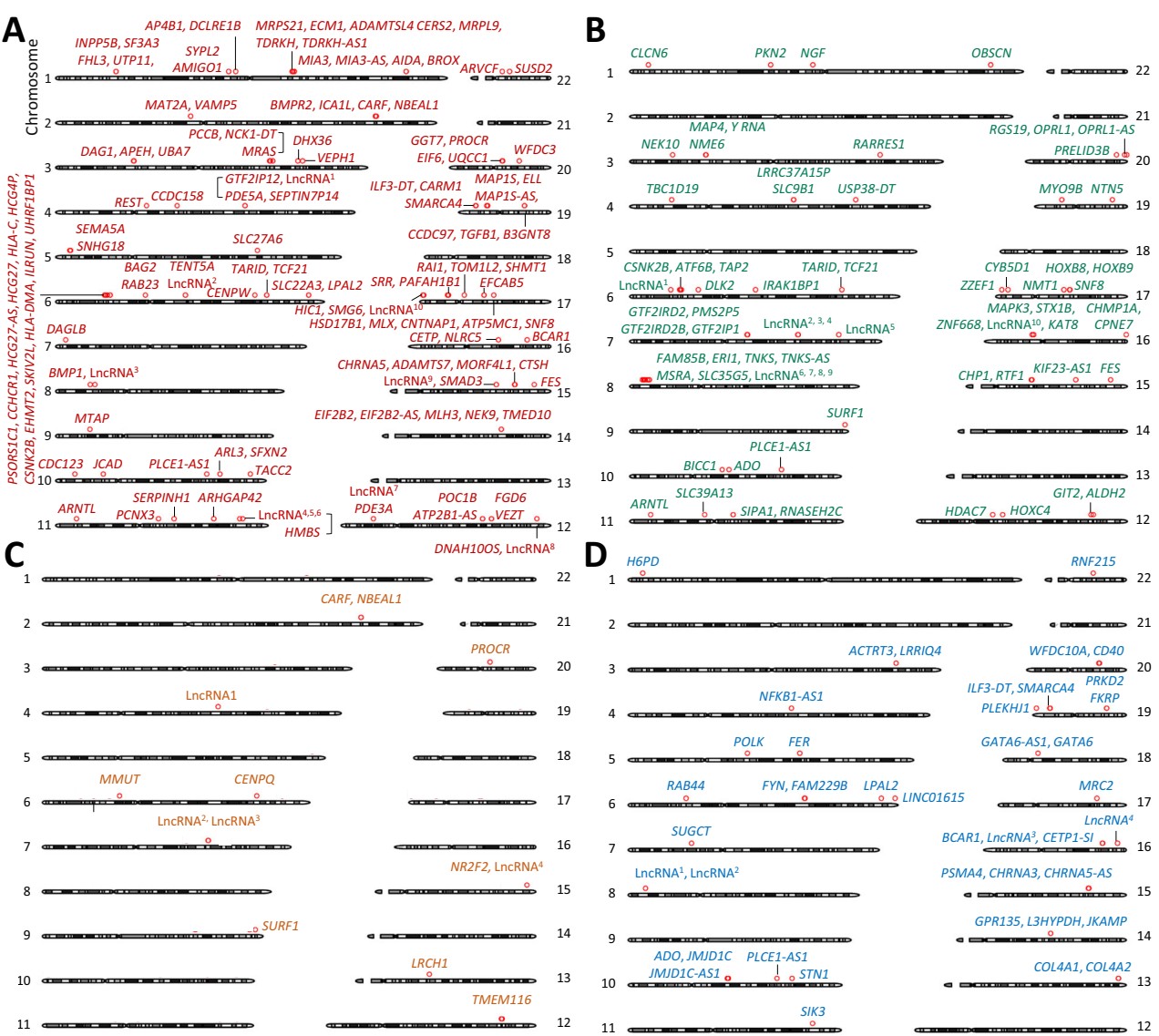

**Fig. 2 | Credible causal genes.** Chromosomal plots of credible causal genes for CAD (**A**), hypertension (**B**), stroke (**C**) and AAA (**D**), respectively. The identities of the long non-coding RNA (lncRNA) genes are described in Supplementary Data 5–8.

Among these potential targets, 38 encode proteins for which there are existing drugs[26] (Fig. 4 and Supplementary Data 17,19,21 & 23).

### Influence of likely causal genes on VSMC behaviour

To assess possible effects of the likely causal genes on VSMC behaviour, we performed high-throughput pooled CRISPR-Cas9-nuclease knockout screens targeting the protein-coding genes among these likely causal genes (this technique is unable to test non-coding RNA genes). The results of these screens suggested that some of these genes might influence VSMC proliferation or migratory ability (Supplementary Data 24, 25, and Fig. 5A, B). A selection of these genes was subjected to validation experiments, with the following selection criteria: (1) among the top 10 genes ranked by score in the pooled CRISPR-Cas9-nuclease knockout screens; (2) considered to be a pleiotropic gene for more than one vascular disease in the aforementioned cross-trait analysis; and (3) likely to be druggable in the abovementioned druggability analysis. In the validation experiments, VSMCs were transfected with either siRNA for the selected gene or negative control siRNA, followed by a VSMC proliferation or migration assay. The following genes met the selection criteria and were tested in the validation experiments: *BCAR1*, *CARF*, and *SMARCA4* for VSMC proliferation, and *CSNK2B* and *FES* for VSMC migration. The validation experiments confirmed the CRISPR-Cas9-nuclease knockout screen findings for *BCAR1*, *CARF*, *FES*, and *SMARCA4*, with the same direction of effect in both the screen and validation experiments (Fig. 5C, D and Supplementary Fig. 9A, B), but not for *CSNK2B* (Supplementary Fig. 9C). For instance, the CRISPR-Cas9-nuclease screen showed that *FES* knockout increased VSMC migration, and this finding was supported in the validation experiment where we observed that siRNA-induced knockdown of *FES* increased the migratory ability of VSMCs (Fig. 5D).

### Effect of FES on VSMC migration and production of matrix metalloproteinases

Following the validation experiments, we carried out further functional experiments with a particular focus on *FES*. As the FES protein has been reported to participate in intracellular signalling[27], we investigated if *FES* knockdown might influence certain signalling

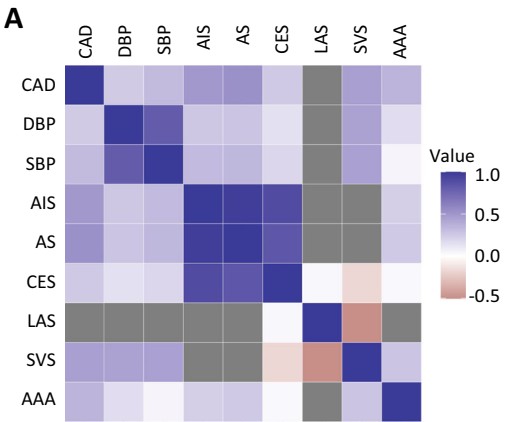

**A**

**B**

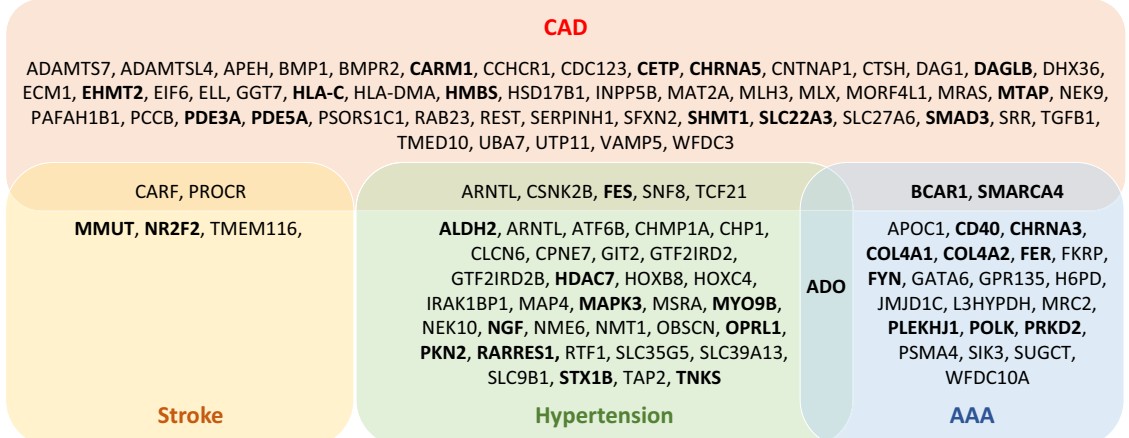

LncRNA[1]: ENSG00000243797; LncRNA[2]: ENSG00000267052.

**Fig. 3 | Genetic correlations and pleiotropic genes. A** Genetic correlations between CAD, blood pressure, hypertension, various types of stroke, and AAA. Abbreviations: *AAA* abdominal aortic aneurysm; *AIS* arterial ischaemic stroke; *AS* all stroke; *CAD* coronary artery disease; *CES* cardioembolic stroke; *DBP* diastolic blood pressure; *LAS* large artery stroke; *SBP* systolic blood pressure; *SVS* small vessel stroke. Grey box indicates data unavailable. **B** Names of candidate causal genes shared by more than one vascular disease.

**CAD**

ADAMTS7, ADAMTSL4, APEH, BMP1, BMPR2, **CARM1**, CCHCR1, CDC123, **CETP**, **CHRNA5**, CNTNAP1, CTSH, DAG1, **DAGLB**, DHX36, ECM1, **EHMT2**, EIF6, ELL, GGT7, **HLA-C**, HLA-DMA, **HMBS**, HSD17B1, INPP5B, MAT2A, MLH3, MLX, MORF4L1, MRAS, **MTAP**, NEK9, PAFAH1B1, PCCB, **PDE3A**, **PDE5A**, PSORS1C1, RAB23, REST, SERPINH1, SFXN2, **SHMT1**, **SLC22A3**, SLC27A6, **SMAD3**, SRR, TGFB1, TMED10, UBA7, UTP11, VAMP5, WFDC3

| CARF, PROCR | ARNTL, CSNK2B, **FES**, SNF8, TCF21 | **BCAR1**, **SMARCA4** |
|---|---|---|
| **MMUT**, **NR2F2**, TMEM116, | **ALDH2**, ARNTL, ATF6B, CHMP1A, CHP1, CLCN6, CPNE7, GIT2, GTF2IRD2, GTF2IRD2B, **HDAC7**, HOXB8, HOXC4, IRAK1BP1, MAP4, **MAPK3**, MSRA, **MYO9B**, NEK10, **NGF**, NME6, NMT1, OBSCN, **OPRL1**, **PKN2**, **RARRES1**, RTF1, SLC35G5, SLC39A13, SLC9B1, **STX1B**, TAP2, **TNKS** | APOC1, **CD40**, **CHRNA3**, **COL4A1**, **COL4A2**, **FER**, FKRP, **FYN**, GATA6, GPR135, H6PD, JMJD1C, L3HYPDH, MRC2, **PLEKHJ1**, **POLK**, **PRKD2**, PSMA4, SIK3, SUGCT, WFDC10A |
| **Stroke** | **Hypertension** | **AAA** |

(with ADO column between Hypertension and AAA)

**Fig. 4 | Druggability of credible causal genes.** The boxes show the names of genes that encode potentially druggable proteins. Bold font indicates genes that encode proteins for which there are existing drugs.

pathways and result in changes in the production of certain proteins. Thus, we performed whole-genome RNA-seq, quantitative proteomics and phosphoproteomics assays on VSMCs with and without siRNA-induced *FES* knockdown. The phosphoproteomics analysis identified a number of proteins whose phosphorylation status might be influenced by *FES* knockdown (Supplementary Data 26) and a pathway analysis of these proteins suggested that they were enriched in several pathways including the epidermal growth factor (EGF)-, VEGF-, and CCKR (cholecystokinin A receptor)-signalling pathways (Fig. 6A). The RNA-seq and quantitative proteomics analyses indicated that *FES* knockdown could potentially influence the production of a number of genes/proteins, including the extracellular matrix protein-degrading enzymes matrix metalloproteinase-1 (MMP1) and MMP3 (Fig. 6B, C and Supplementary Data 27–29). Reverse transcription-polymerase chain reaction (RT-PCR) and Western blot analyses confirmed increased MMP1 and MMP3 production in VSMCs with *FES* knockdown (Fig. 7A–C) and additionally showed that *FES* knockdown decreased the levels of contractile VSMC phenotype markers (Fig. 7D, E).

### Effect of FES on atherosclerosis development and blood pressure in mouse models

In the colocalization analysis described early, *FES* was identified as a pleiotropic gene for both CAD and hypertension.

We previously showed that *Fes*[-/-]/*Apoe*[-/-] mice had larger aortic-root atherosclerotic lesions with a higher content of VSMCs and macrophages, compared with *Fes*[+/+]/*Apoe*[-/-] littermates, after a high-fat diet for 12 weeks[28]. To further assess the effect of *Fes* on atherogenesis, we performed *en face* Oil Red O staining on aortae taken from the *Fes*[-/-]/*Apoe*[-/-] mice and their *Fes*[+/+]/*Apoe*[-/-] littermates, and found that *Fes*[-/-]/*Apoe*[-/-] mice had significantly larger areas of atherosclerotic lesions than *Fes*[+/+]/*Apoe*[-/-] littermates (mean ± standard deviation: 8.34 ± 2.54 versus 6.06 ± 2.35; $P = 0.013$) (Fig. 8A), which was observed in male mice but not in female mice when data from male and female mice were analysed separately (Supplementary Fig. 10).

To assess the impact of *FES* on blood pressure, we performed hemodynamic measurements in unconscious *Fes*[-/-]/*Apoe*[-/-] mice and their *Fes*[+/+]/*Apoe*[-/-] littermates, both of which had been on a chow diet. At baseline, *Fes*[-/-]/*Apoe*[-/-] mice had a significant elevation in both systolic (88.0 ± 10.1 mmHg versus 104.4 ± 6.7 mmHg, *Fes*[+/+]/*Apoe*[-/-] versus *Fes*[-/-]/*Apoe*[-/-], $P = 0.042$) and diastolic (58.8 ± 8.9 mmHg versus 74.9 ± 9.0 mmHg, *Fes*[+/+]/*Apoe*[-/-] versus *Fes*[-/-]/*Apoe*[-/-], $P = 0.042$) blood pressure (Fig. 8B). The increase in blood pressure in *Fes*[-/-]/*Apoe*[-/-] mice did not lead to a compensatory change in heart rate, with no difference observed in heart rate at baseline between the two groups (Supplementary Fig. 11A). As expected, acetylcholine caused a reduction in both systolic and diastolic blood pressure in both groups. However, in

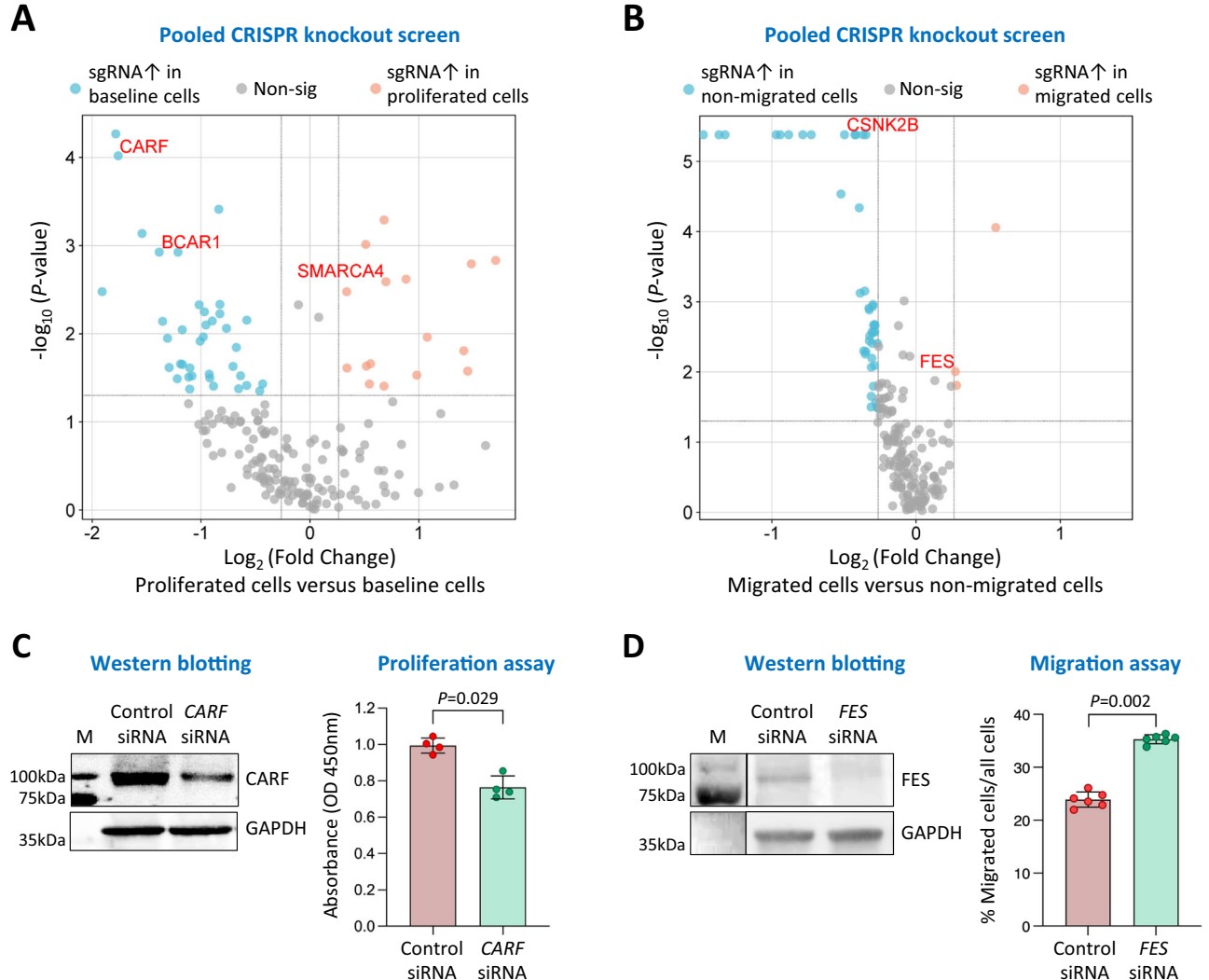

**Fig. 5 | CAD/hypertension/stroke/AAA credible causal genes that affect VSMC proliferation or migration. A**, **B** Pooled CRISPR screens of CAD/hypertension/stroke/AAA credible causal genes that affect VSMC proliferation (**A**) or migration (**B**). Pink symbols indicate genes whose knockout promotes proliferation (**A**) or migration (**B**); blue symbols indicate genes whose knockout reduces proliferation (**A**) or migration (**B**). Shown in the y-axis are $\log_{10}$ transformed nominal *P*-values from a two-sided permutation test. **C** Results of Western blot analysis of CARF (left) and proliferation assay (right) of VSMCs transfected with either CARF siRNA or control siRNA. OD: Optical density. **D** Results of Western blot analysis (left) and migration assay (right) of VSMCs transfected with either FES siRNA or control siRNA. Charts in (**C**, **D**) show mean ± standard deviation (error bars); $n = 4$ (**C**) or 6 (**D**) technical replicates; *P*-values from a two-tailed Mann-Whitney test; source data are provided as a Source Data file. M, molecular weight marker.

$Fes^{-/-}/Apoe^{-/-}$ mice this response was significantly blunted with systolic blood pressure reduced by only $14.6 \pm 3.8$ mmHg in $Fes^{-/-}/Apoe^{-/-}$ mice versus $22.2 \pm 4.3$ mmHg in $Fes^{+/+}/Apoe^{-/-}$ mice ($P = 0.027$) (Fig. 8C). A similar result was observed in diastolic responses with a greater reduction in diastolic blood pressure observed in $Fes^{+/+}/Apoe^{-/-}$ mice compared with $Fes^{-/-}/Apoe^{-/-}$ mice ($P = 0.014$) (Fig. 8C). In contrast, no difference was observed in either systolic or diastolic blood pressure between the two groups in response to phenylephrine (Supplementary Fig. 11B).

**Association of FES gene mutations with susceptibility to CAD and hypertension in humans**

In addition to the above mouse model experiments, through the AstraZeneca PheWAS Portal[25], we collated human data of deleterious genetic variants in the protein-coding regions of the human *FES* gene in relation to blood pressure, hypertension, and the CAD traits angina and myocardial infarction, based on whole genome sequencing and phenotypic measurements of European participants in the UK Biobank. Collapsing analyses of *FES* protein truncating variants showed that, collectively, they were significantly associated with elevated systolic and diastolic blood pressure ($P = 4.70 \times 10^{-5}$ for systolic blood pressure and $2.70 \times 10^{-4}$ for diastolic blood pressure)(Table 1), and a higher likelihood of hypertension [odds ratio (95% confidence interval) = 1.93 (1.30–2.86)](Table 2), as well as a > 2-fold increased risk of myocardial infarction [2.30 (1.19–4.44)] and angina [2.05 (1.16–3.61)][25] (Table 2).

## Discussion

To facilitate medical translation from the recent discovery of the many genomic loci associated with CAD, hypertension, stroke and/or AAA, it is important to identify the causal genes at these loci to better understand the disease pathogenesis and utilise such information to identify new targets for therapeutic development. Our study, focused on VSMCs, a key cell type involved in the pathogenesis of these diseases, was aimed at this purpose and provides important findings in several aspects.

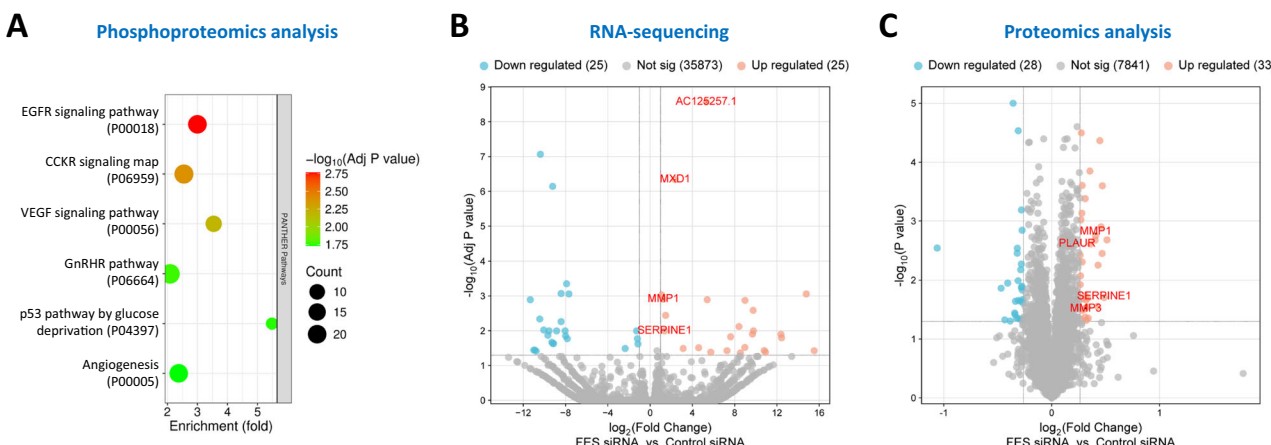

**Fig. 6 | Results of phosphoproteomics, RNA-sequencing, and quantitative proteomics analyses of vascular smooth muscle cells transfected with either FES siRNA or control siRNA. A** Pathways enriched with phosphorylated proteins in vascular smooth muscle cells (VSMCs), transfected with FES siRNA versus VSMCs transfected with control siRNA. CCKR: cholecystokinin A receptor; EGFR: epidermal growth factor receptor; GnRHR: gonadotropin-releasing hormone receptor; VEGF: vascular endothelial growth factor. **B** Up- or down-regulated genes in VSMCs transfected with FES siRNA versus VSMCs transfected with control siRNA. **C** Up- or down-regulated proteins in VSMCs transfected with FES siRNA versus VSMCs transfected with control siRNA. *P*-values from two-sided Fisher's Exact (**A**) or two-sided Student's *t* test (**B**, **C**).

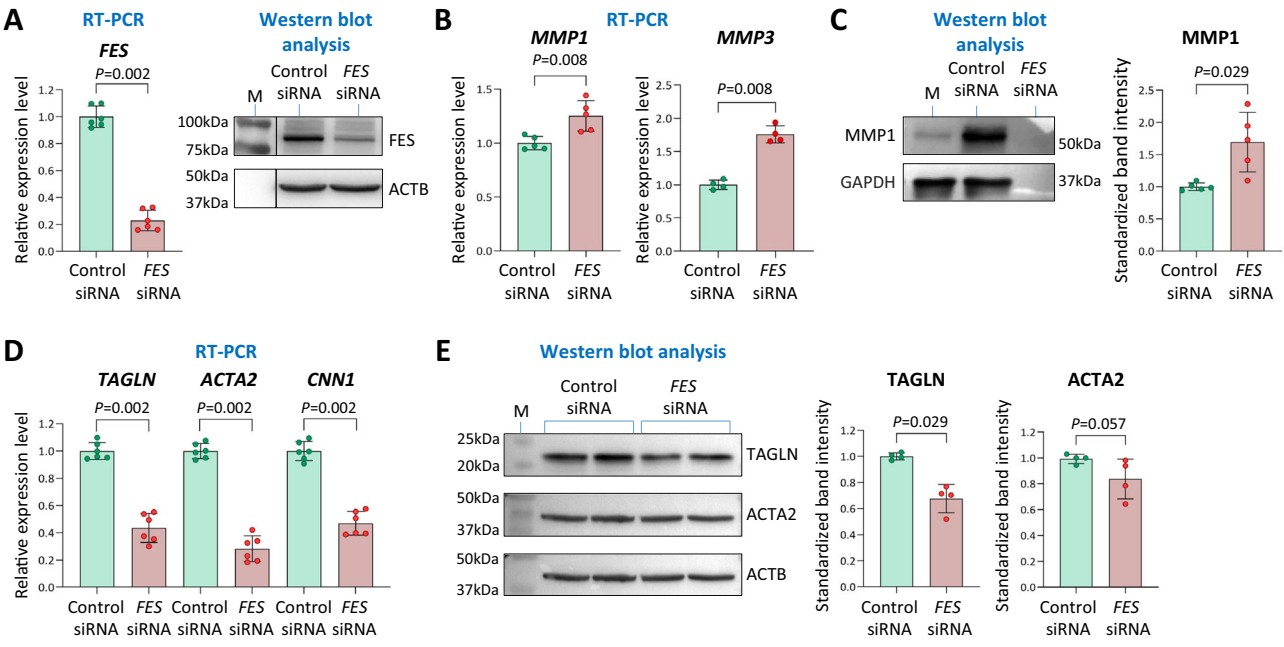

**Fig. 7 | Effect of FES on the levels of MMPs and contractile VSMC markers. A** Reverse transcriptase polymerase chain reaction (RT-PCR) and Western blot analyses confirming FES knockdown in VSMCs transfected with *FES* siRNA compared with VSMCs transfected with control siRNA. **B** RT-PCR analysis of matrix metalloproteinase 1 (*MMP1*), *MMP2*, and *MMP3*. **C** Western blot analysis of MMP1. **D** RT-PCR analysis of the contractile VSMC phenotype markers *ACTA2*, *TAGLN*, and *CNN1*. **E** Western blot analysis of ACTA2 and TAGLN. **A, C, E** Average representative Western blot images are shown. M: Protein size markers; ACTB: β-actin. **A–E** Column charts show mean ± standard deviation (error bars) values; *n* = 4 (**C, E**), 5 (**B**), or 6 (**A, D**) technical replicates; *P*-values shown are from two-tailed Mann-Whitney test; source data are provided as a Source Data file.

First, using an integrative functional genomics approach to systematically analyse regulatory genetic variants and their associated genes in conjunction with disease GWAS data, our study was able to identify panels of genes that are likely to be causal at the loci associated with CAD, hypertension, stroke, and/or AAA. Reassuringly, some of these genes have been demonstrated in previous studies using other methods to be important players in relevant disease processes, in particular, *ADAMTS7*[29], *JCAD*[30], *SMAD3*[31] and *TCF21*[17] in atherogenesis; *ARNTL*[32], *CLCN6*[33] and *PKN2*[34] in blood pressure regulation, and *COL4A1/COL4A2*[35] and *SMARCA4*[36] in aortic aneurysm development.

This indicates the robustness of our analyses and provides further evidence supporting the hypothesis that these genes have a pathological role. Importantly, our analyses additionally identified multiple other genes that are likely to be causal but have hitherto not been implicated in the vascular diseases under study. These findings provide insight into the genetically mediated pathogenesis of these diseases.

The functional pathway enrichment analysis of the likely causal genes identified highlights TGFβ signalling as a key pathway mediating the genetic influence on CAD susceptibility, substantiating findings from previous studies[15,16,37]. TGFβ is known to be a key regulator of

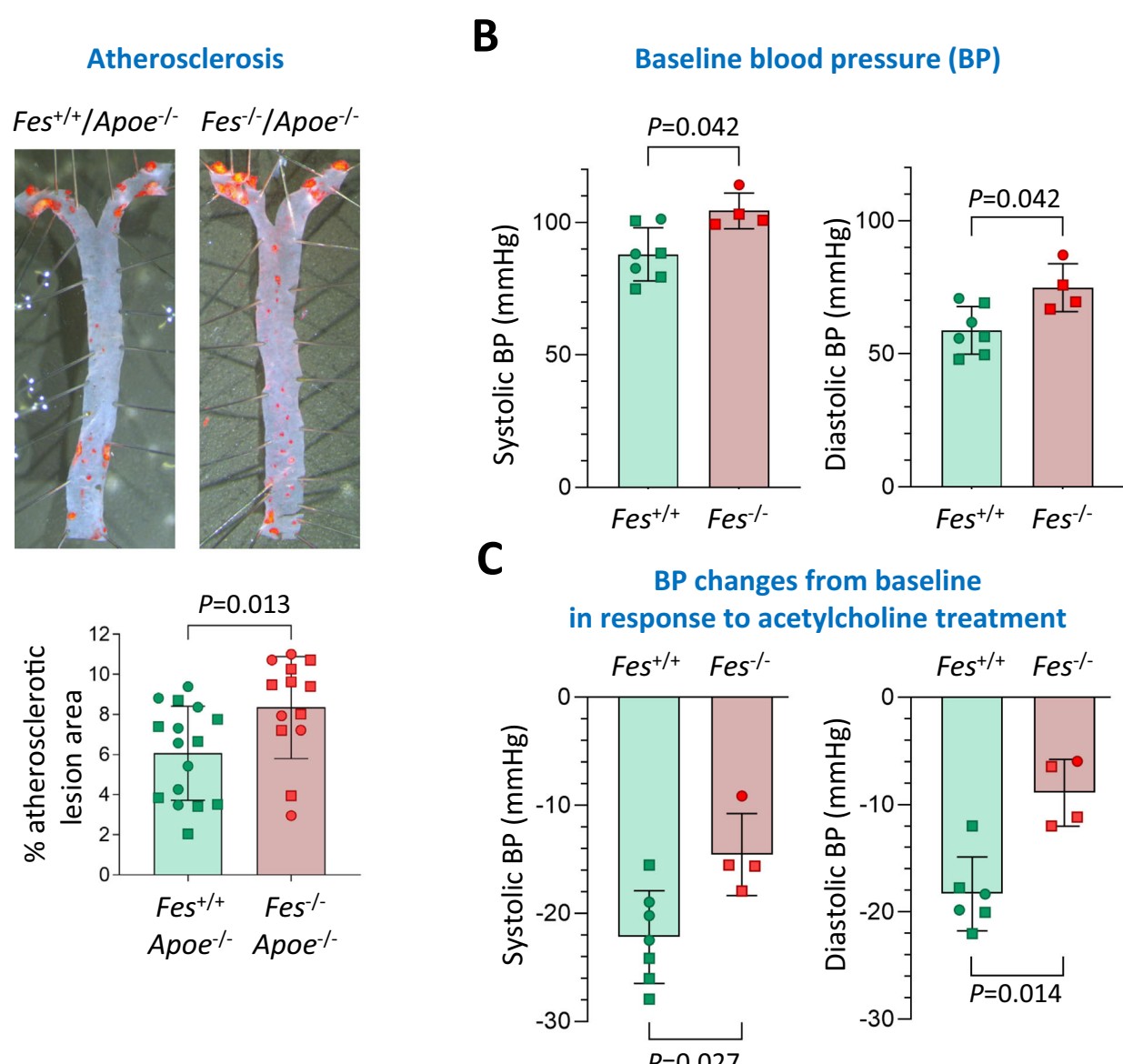

**Fig. 8 | Atherosclerotic lesions and blood pressure in Fes$^{+/+}$ mice and Fes$^{-/-}$ mice.**
**A** Results of *en face* Oil-red-O staining of aortae. Representative images are shown.
**B** Baseline blood pressure. **C** Blood pressure changes from baseline in response to acetylcholine treatment. **A**–**C** In the column charts, square symbols represent values from male mice whilst circle symbols represent values from female mice; columns and error bars show mean ± standard deviation; *P*-values are from a two-sided Mann-Whitney test; $n = 13$–16 mice/group in (**A**) and $n = 4$–7 mice/group in B&C; source data are provided as a Source Data file.

**Table 1 | Association of FES protein truncating variants with blood pressure in UK Biobank subjects**

| Phenotype | Subjects | Subjects with variants | Effect size | *P* -value |
|---|---|---|---|---|
| Systolic BP | $n = 431934$ | $n = 141$ (0.033%) | 0.32 (0.16–0.47) | $4.70 \times 10^{-5}$ |
| Diastolic BP | $n = 431938$ | $n = 141$ (0.033%) | 0.30 (0.14–0.46) | $2.70 \times 10^{-4}$ |

*P*-values were from two-sided linear regression analysis.

VSMC proliferation[38]. In addition, the analysis underlines a genetic influence on the VEGF signalling pathway in hypertension, which is likely to be related to the role of VEGF in regulating peripheral vascular resistance[39], because the VEGF inhibitor pazopanib has been reported to raise blood pressure with increased peripheral vascular resistance[40]. Furthermore, the analysis indicates that several pathways (through VEGFR, PDGFR, integrin and cadherin, respectively) known to influence VSMC proliferation, VSMC migration and cell-matrix interaction[41-43], are at play in AAA development.

Another notable finding of our study was the significant genetic correlations between CAD, blood pressure, stroke and AAA, suggesting that these diseases share considerable genetic underpinnings. This finding at the genetic level is in accordance with the fact that some common cellular mechanisms (such as altered VSMC behaviour) and pathological changes (such as atherosclerosis) are involved in the pathogenesis of several of these vascular diseases[39]. Importantly, we identified 18 pleiotropic genes which are likely to be involved in the development of more than one of these diseases.

**Table 2 | Association of FES protein truncating variants with hypertension, angina, and myocardial infarction in UK Biobank subjects**

| Phenotype | Cases | Cases with variants | Controls | Controls with variants | Odds ratio (95% CI) | *P*-value |
|---|---|---|---|---|---|---|
| Hypertension | $n = 188129$ | $n = 86$ (0.046%) | $n = 147753$ | $n = 35$ (0.024%) | 1.93 (1.30–2.86) | 0.001 |
| Angina | $n = 34976$ | $n = 14$ (0.054%) | $n = 120611$ | $n = 32$ (0.026%) | 2.05 (1.16–3.61) | 0.018 |
| Myocardial infarction | $n = 26883$ | $n = 14$ (0.052%) | $n = 105893$ | $n = 24$ (0.023%) | 2.30 (1.19–4.44) | 0.015 |

*P*-values were from a two-sided logistic regression analysis.

In support, we observed that mice with knockout of *FES*, one of the pleiotropic genes, had increased atherosclerosis as well as raised blood pressure, and additionally gathered evidence of *FES* gene deleterious variants significantly increasing the risk of both hypertension and CAD in humans among UK Biobank participants. Our experiments suggested that FES knockdown in VSMCs could influence the EGF-, VEGF-, and CCKR-signalling pathways, reduce contractile VSMC phenotype markers, promote VSMC migration, and increase production of the proteases MMP1 and MMP3. These findings are in agreement with previous reports that FES can interact with the EGF receptor[44,45], that EGF induces transition of VSMCs from a differentiated/contractile state to a dedifferentiated/synthetic phenotype[46], and that EGF-, VEGF-, and CCKR-signalling upregulates MMP expression[41,47–50].

Taken together, these findings suggest that FES can interact with these signalling pathways, whilst FES reduction can induce VSMC dedifferentiation to a synthetic phenotype with increased mobility and MMP expression. It has long been documented and is recently re-emphasised[51,52] that VSMC dedifferentiation and their accumulation significantly contribute to atherosclerotic plaque formation and enlargement, although VSMCs in the fibrotic cap are beneficial for plaque stability. Furthermore, both MMP1 and MMP3 are capable of degrading a variety of extracellular proteins in the vascular wall and have been shown to facilitate VSMC migration, matrix degradation and vascular remodelling[53]. These processes are instrumental in the pathogeneses of both atherosclerosis and hypertension[53–55].

Recent studies have indicated that using genetic data to nominate therapeutic targets can significantly increase the likelihood of success in the development of new drugs[56–58]. In this context, another notable finding of our study is the identification of a panel of potential targets that are known or expected to be druggable. Remarkably, several of the pleiotropic genes identified are potentially druggable, raising the possibility of developing drugs to target these genes for the treatment of multiple co-existing vascular diseases.

We note some limitations of our work. First, our study used human umbilical cord-derived VSMCs for eQTL mapping in the context of adult cardiovascular disease. At present, a large collection of adult VSMCs is unavailable. Therefore, in this study, we utilised a large collection of human umbilical cord-derived VSMCs (VSMC eQTL data from 1486 individuals) to provide high statistical power for the comprehensive genome-wide eQTL analysis. As the cells were isolated from umbilical cords from individuals of the same biological age, the analysis also benefited from circumventing age-related confounding. However, there is a caveat that neonatal cells can differ from adult cells in gene regulation and expression. Notwithstanding, in a recent work[16], we compared the transcriptomes of the umbilical cord-derived VSMCs utilised in this study with reported transcriptomic data from other types of cell/tissue[59,60], and found that the gene expression profile of the human umbilical cord-derived VSMCs mapped closer to adult human coronary artery smooth muscle cells than any other cell/tissue types[16]. Second, as *FES* is expressed in not only VSMCs but also some other cell types such as monocytes and endothelial cells, the total atherogenic contribution of *FES* is expected to derive from its effect on the various cell types, including VSMCs, as suggested here and monocytes as indicated in our recent study[28]. The relative contributions from the different cell types are yet to be established.

Notwithstanding, the primary aim of our present study is not to test a cell type-specific effect on vascular diseases, but to identify potential causal genes at the genomic loci that have been found to be associated with CAD, hypertension, stroke and/or AAA in GWAS. Third, the in vivo experiments of *Fes* knockout were performed on small numbers of mice and were not powered to test possible sex-specific effects of *Fes* on atherosclerosis and blood pressure. Fourth, the CRISPR knockout screens and follow-up experiments were carried out on VSMCs from a single donor and so could not account for potential inter-individual biological variations. Therefore, the findings from these experiments are exploratory and for hypothesis generation, rather than definitive.

In summary, the findings from this systematic, integrative functional genomics study provide a better understanding of the genetic underpinnings and treatment targets of several major vascular diseases.

## Methods
### Generation of a VSMC eQTL catalogue
The study was approved by the East Midlands - Derby Research Ethics Committee (20/EM/0028), and the parents of the umbilical cord donors gave written informed consent.

The collection of VSMCs from different individuals and the methods of genotyping, imputation, RNA-Seq, eQTL analysis, and VSMC behaviour assays have been described previously[16]. In brief, VSMCs were isolated from the artery of umbilical cords from 2114 different donors, with the use of a well-established method[61]. DNA from 1992 samples was genotyped for 760,000 variants using the Infinium Global Screening Array-24 2.0 BeadChips (Illumina), and imputation was subsequently performed to obtain genotypic information of 7,334,165 variants. RNA-seq was conducted on total RNA extracted from passage 3 VSMCs from 1499 donors. For RNA-seq, a strand-specific library with rRNA removal was prepared from total RNA of each of the 1499 VSMC samples, and 150 bp paired-end sequencing, at a read depth of 30 million, was performed using the Illumina platform. An eQTL analysis was carried out on 1486 VSMC samples that had both genotype and RNA-Seq data, applying the eigenMT-BH method[62] for multiple testing correction.

### Compilation of tagging SNPs that marked the genetic variants/haplotypes reported to be associated with CAD, blood pressure/hypertension, stroke, and AAA, respectively
We previously identified 424 tagging SNPs which marked the genetic variants/haplotypes that had been reported to be associated with CAD at the genome-wide significance level ($P < 5 \times 10^{-8}$) in GWAS[1,3,4,63,64], did not have significant linkage disequilibrium (LD) between them, and were typed in the abovementioned VSMC collection either directly by using the Global Screening Array or indirectly by imputation[16]. In the present study, we further compiled lists of tagging SNPs which marked the genetic variants/haplotypes that had been reported to be associated with blood pressure/hypertension[5,19], stroke[5–7], and AAA[5,8], respectively, at the genome-wide significance level ($P < 5 \times 10^{-8}$) in GWAS, did not have significant LD between them, and were typed in the abovementioned VSMC collection either directly by using the Global Screening Array or indirectly by imputation[16]. LD between the SNPs was assessed using LDlink's SNPclip programme, which removed

SNPs based on LD in a given population and kept the first SNP (based on genomic position). We used an $R^2$ threshold of 0.8 and an MAF threshold of 0 with the "European" option. If there was available LD data, SNPs were included/excluded as required, but if not, the SNPs were kept in the list and assumed to be independent signals. In total, 1402 independent SNPs were identified for blood pressure/hypertension (Supplementary Data 30), 139 for stroke (Supplementary Data 31), and 157 for AAA (Supplementary Data 32). Of these, 1203 for blood pressure/hypertension, 119 for stroke, and 136 for AAA, were available in this study (either directly or via proxy SNPs with $R^2 > 0.8$), which were typed in the abovementioned VSMC collection[16] by the Global Screening Array or imputed, and were not filtered out (missingness < 0.05, MAF > 0.01; HWE $P$-value > $1 \times 10^{-6}$).

## Analyses of colocalization between disease GWAS and VSMC eQTL signals

**Source and quality control of CAD GWAS datasets.** CAD GWAS summary statistics from Nelson et al.[2], van der Harst and Verwei[3], Howson et al.[4], and Aragam et al.[1] were downloaded from http://www.cardiogramplusc4d.org (accessed on February 18, 2021), ftp://ftp.ebi.ac.uk/pub/databases/gwas/summary_statistics/GCST005001-GCST006000/GCST005194 (accessed on February 18, 2021), https://graspnhlbinihgov.ezproxy3.lib.le.ac.uk/FullResults.aspx (accessed on March 29, 2021), and http://ftp.ebi.ac.uk/pub/databases/gwas/summary_statistics/GCST90132001-GCST90133000/GCST90132314 (accessed December 13, 2022), respectively. Blood pressure GWAS summary statistics from Evangelou et al.[19] (GCST006624, GCST006630) were downloaded from http://ftp.ebi.ac.uk/pub/databases/gwas/summary_statistics/GCST006001-GCST007000 (accessed on February 02, 2022), and hypertension GWAS summary statistics from Neale[5] were downloaded from http://www.nealelab.is/uk-biobank. Stroke GWAS summary statistics from Malik et al.[6] and Mishra et al.[7] (GCST006909, GCST006907, GCST006910, GCST006906, GCST90104543, GCST90104542, GCST90104541, GCST90104539) were downloaded from GWAS catalogue https://www.ebi.ac.uk/gwas/ (accessed April 02, 2022 and December 13, 2022), and AAA GWAS summary statistics from Roychowdhury et al.[8] were obtained from the study authors, respectively.

Cleaning and quality control steps were applied to the raw summary statistics data before they were used. Genome coordinates were converted to the hg38 build with CrossMap[65] and SNP marker names were updated with the new coordinates. If absent, variance, z-scores and minor allele frequencies were calculated in R v3.6.1 (https://www.R-project.org).

**eCAVIAR analyses.** We used eCAVIAR[20] to test for colocalization of disease GWAS with VSMC *cis*-eQTL. The eCAVIAR analysis was performed on SNPs that had been reported to associate with CAD[1–4], hypertension[5], stroke[6,7] or AAA[8], and QTL data from the abovementioned VSMC eQTL catalogue. The top eQTL SNP (i.e., SNPs with the lowest $P$-value) of genes were selected and SNPs within 500 kb of the top SNP were overlapped with GWAS summary data. We analysed 500 kb windows as in several other studies[16,59,66], to include most, if not all, SNPs in LD with the top SNPs, since significant LD between SNPs typically decays within 100 kb in outbred human populations[67–69]. Supplementary Fig. 12 shows a locus zoom plot of a CAD-associated locus on chromosome 15, at which all CAD-associated SNPs are located within 100 kb. 500 kb windows were more extensive than 200 kb windows used in the originally reported eCAVIAR method[20]; nevertheless, fixed windows of any size have the limitation of not accounting for between-region differences in LD structure. The eCAVIAR colocalization test was performed for loci that had 5 or > 5 overlapping SNPs present in both the eQTL and GWAS datasets, to reduce potential false positives due to overlaps by chance. We estimated LD for overlapping GWAS SNPs from individuals of European ancestry in the 1000 Genomes Phase 3 using PLINK[70]. LD of *cis*-eQTL overlapping SNPs was estimated from VSMC genotype data. The number of causal SNPs was set to be up to 2 per locus, and colocalization events with > 0.05 colocalization posterior probability (CLPP) were considered significant.

The above analyses were performed on SNPs reported to be associated with CAD[1–4], hypertension[5], or AAA[8], at the genome-wide significant level ($P < 5 \times 10^{-8}$), and on SNPs reported to be associated with stroke[6,7] at $P < 5 \times 10^{-5}$ as the number of SNPs reported to be associated with stroke at $P < 5 \times 10^{-8}$ was small. However, only the SNPs with both CLPP > 0.05 in eCAVIAR and $P < 5 \times 10^{-8}$ in stroke GWAS[6,7] were considered to be likely causal SNPs for stroke.

**SMR/HEIDI analyses.** We also used SMR/HEIDI[21] to test for colocalization between disease GWAS and VSMC *cis*-eQTL. We performed SMR/HEIDI analyses with default settings. LD was estimated with individuals of European ancestry in the 1000 Genomes Phase 3 as reference. Although we ran transcriptome-wide SMR/HEIDI analysis, we focused only on SNPs that had been reported to associate with CAD[1–4], hypertension[5], stroke[6,7], or AAA[8], at the genome-wide significance level ($P$-value < $5 \times 10^{-8}$) in GWAS. We applied a threshold of $p_{SMR} < 0.05$ & $p_{HEIDI} > 0.05$ to identify significant colocalizations of CAD GWAS signals with VSMC eQTL signals.

## Analyses of colocalization between disease GWAS and arterial tissue eQTL signals

With the use of LocusCompare V2 (https://www.locuscompare2.com/) and its default settings, additional colocalization analyses were performed on the abovementioned summary statistics of GWASs of CAD[1–4], hypertension[5], stroke[6,7], and AAA[8], respectively, in conjunction with eQTL data from coronary artery tissues and aortic artery tissues, respectively, from the GTEx Portal. eCAVIAR CLPP > 0.05, SMR $P$-value < $10^{-6}$, Coloc (H4) > 0.8, PrediXcan $P$-value < $10^{-6}$, fastEnloc GLCP > 0.05, and/or FUSION $P$-value < $10^{-6}$, were considered significant.

## ATAC-seq

ATAC-seq was performed on 3 biological replicates of VSMCs derived from umbilical arteries from different individuals, as described previously[71]. In brief, $5 \times 10^4$ VSMCs were lysed with cold lysis buffer, and then the nuclei were incubated with a transposition reaction buffer containing Tn5 transposase (Illumina, FC-121-1030) for 30 minutes at 37 °C. The resulting libraries were subjected to polymerase chain reaction (PCR) with high-fidelity polymerase (New England Labs, M0541) and Nextera barcodes. The libraries were then multiplexed and sequenced using the Novaseq 6000 SP Flow Cell (Illumina) with > 100 million reads (50 bp pair end) per individual indexed library.

ATAC-seq data were processed using Galaxy (https://usegalaxy.eu and https://usegalaxy.org)[72,73]. In brief, the quality of the raw data was evaluated with FastQC (v0.11.9). Adaptors were removed using trimmomatic (v0.38), and paired-end reads aligned to the human genome (hg38) using Bowtie2 (v2.4.2) with the --very-sensitive, --dovetail and --X 1000 maximum insert size parameters. Mitochondrial and duplicate reads were removed using bamtools (v2.4.0) and Picard MarkDuplicates (v2.18.2), resulting in an average of 45 million filtered reads per sample for downstream analyses. Peaks were called using MACS2 (v2.1.1.20160309) with the start sites of the reads extending by 200 bp (100 bp in each direction) to assess coverage and a P-value 0.05 cutoff to reveal open chromatin peaks. The intersect function of BedTools v2.30.0 was used to identify overlaps of MACS2 peaks with the chromosome 7p15.3 and 7q23.3 loci at which variants were found to be associated with VSMC apoptosis/ACD in this study. Pygenometracks (v3.6) was used for visualisation of genomic data tracks with publicly available ENCODE ChIP-seq SMC embryo-origin datasets (ENCSR116JEF: H3K27ac experiment ENCSR210ZPC, H3K4me3

experiment ENCSR515PKY and H3K27me3 experiment ENCSR143RMH) and the generation of MACS2 peak and coverage files.

## H3K27ac HiChIP-seq

H3K27ac HiChIP-seq was performed as described previously[74,75], with some modifications. Two independent H3K27ac ChIP-seq on human coronary artery smooth muscle cells ($>1 \times 10^7$) (ThermoFisher Scientific, C-017-5C) were performed, prior to enriching ligated junctions. Briefly, cells were crosslinked with 1% formaldehyde. Hi-C was performed with the use of the Arima High Coverage HiC Kit (Arima Genomics, A410110) according to the manufacturer's protocol. Thereafter, sheared chromatin samples were subjected to immuno-precipitation by an incubation with 5 μg ChIP-grade anti-histone H3 (acetyl K27) antibody (Abcam, ab4729) and 50 μl Dynabeads Protein G for Immunoprecipitation (ThermoFisher Scientific, 10004D) overnight at 4 °C. After washing, the immunoprecipitated DNA was de-crosslinked at 67 °C for 2 h, eluted in a solution containing 50 mM sodium bicarbonate and 1% sodium dodecyl sulphate, and enriched with the use of Dynabeads MyOne Streptavidin C1 (ThermoFisher Scientific, 65002). Thereafter, a library was prepared with the use of NEBNext Ultra II DNA Library Prep Kit for Illumina (New England Bio-labs, E7645L) according to the manufacturer's protocol. The library was subjected to PCR amplification (12 cycles) with indexed primers, and subsequently amplicons of 300-500 bp in length were selectively collected with the use of Omega Mag-Bind TotalPure NGS (Omega BIO-TEK, M1378). The library was subjected to next-generation sequencing at >300 million paired-end, $2 \times 151$ bp reads, on an illumina HiSeq 4000 platform.

Raw Fastq format paired-end reads were processed with the HiChIP pipeline by Dovertail Genomics (https://hichip.readthedocs.io/en/latest/index.html). The hg38 human genome index generated with Burrows-Wheeler Aligner (http://arxiv.org/abs/1303.3997) was used as reference for alignments and valid pair selection. Duplicate pairs were removed using pairtools[76]. FitHiChIP4 (version 11.0) was used to call significant HiChIP loops at 2.5, 5, 10, 15, and 20Kb resolution. Using ChIP-Seq peak files as input, and initial paired interaction files, high-confidence peak-to-all interactions were processed with the following parameters: loose background (UseP2PBackgrnd = 0), coverage bias regression (BiasType = 1), merge filtering (MergeInt = 1), and 1% FDR (QVALUE = 0.01). High-confidence chromatin interactions were sorted and indexed in bed format, and plotted in R with Givz[77] (version 3.20).

## DNA methylation analysis

DNA methylation assay was performed on 3 biological replicates of human umbilical artery smooth muscle cells (HUASMC, from different individuals) and on human aortic smooth muscle cells (HAoSMC, from 1 individual). Genomic DNA was extracted from the studied cells using the Sigma GenElute Mammalian DNA kit (Sigma Aldrich, G1N70). The DNA samples were subjected to analysis of methylation using the Illumina Infinium MethylationEPIC array. The resulting IDAT files were processed to derive CpG methylation beta values from the array data using the EPIC methylation array-specific ChAMP pipeline[78,79].

The hg38 genomic positions of the CpGs in the Infinium MethylationEPIC array were identified, and the mean methylation beta value (taken from 3 HUASMC and 1 HAoSMC cell lines) was calculated for each CpG. A CpG was defined as having low methylation levels if the mean beta value was $\leq 0.25$. The distance between each CpG and the likely causal variant identified from our colocalization analyses was calculated, and CpG-variant pairs with an absolute distance of $\leq 200$ bp were considered to be within close proximity of each other (for variants with more than one CpG within close proximity, the nearest CpG was selected). Of the close-proximity CpG-variant pairs identified, the proportion that had low methylation (defined above) was calculated (both separately for each vascular disease and combined).

## Pathway analyses

The likely causal genes identified in the colocalization analyses were subjected to gene ontology (GO) enrichment analysis[80]. For each disease, the names of all of the likely causal genes were entered in the Gene Ontology browser (http://geneontology.org/) and searched for various GO aspects, including molecular function, biological process, cellular component, and PANTHER pathways, with the default setting for overrepresentation analysis by Fisher's exact test and with false discovery rate correction. For each GO aspect, the 3 most significant (with the smallest P-values and smallest false discovery rates) GO terms are presented as enrichment bubble plots, which were drawn using the SRplot tool[81].

## Cross-trait LD score regression analysis

We assessed the genetic correlation between each of the cardiovascular diseases and traits using default settings of the software package LDSC (v1.0.1)[82]. The pairwise product of the SNP z statistic was regressed against LD scores of Europeans available in the LDSC software.

## Investigation of phenotypic effects of deleterious protein-coding variants in candidate causal genes

To investigate if deleterious variants in the protein-coding regions of the likely causal genes identified in the colocalization analyses were associated with the disease of interest, we interrogated the candidate genes in the AstraZeneca PheWAS Portal[25] (https://azphewas.com/) (UK Biobank 500k WGS (v2) Public) and PheWeb (https://r8.finngen.fi/). Each of these genes was individually interrogated in both the AstraZeneca PheWAS Portal and PheWeb, and variant-level results were recorded. The results were then filtered for genetic variants associated with the vascular disease of interest, at the genome-wide significance level ($P < 1 \times 10^{-8.7}$ in PheWAS and $P < 5 \times 10^{-8}$ in PheWeb), including traits of CAD (atherosclerotic heart disease, coronary atherosclerosis, ischaemic heart disease, angina pectoris, myocardial infarction, major coronary heart disease event, coronary revascularization, coronary angioplasty, coronary artery bypass grafting), blood pressure/hypertension (systolic blood pressure, diastolic blood press, hypertension, hypertension diseases), stroke, and/or AAA.

To investigate the potential functional consequences of the disease-associated variants identified in the abovementioned search in the AstraZeneca PheWAS Portal and PheWeb, we looked up the variants in the Ensembl Genome Browser 110 (www.ensembl.org) to gather information on functional consequence predictions by SIFT (sorts intolerant from tolerant substitutions)[83], PolyPhen (polymorphism phenotyping)[84], CADD (combined annotation dependent depletion)[85], REVEL (rare exome variant ensemble learner)[86], MetaLR[87], and Mutation Assessor[88]. Each variant was individually interrogated in the Ensembl Genome Browser, and the scores of SIFT, PolyPhen, CADD, REVEL, MetaLR and Mutation Assessor were recorded. A SIFT score $\leq 0.05$, a PolyPhen score $\geq 0.85$, a CADD score $\geq 30$, a REVEL score $\geq 0.75$, and/or a MetaLR $\geq 3.5$ score was considered to indicate that the variant was deleterious.

## Druggability and drug-gene interaction analyses

To investigate whether any of the candidate causal genes identified in this study were potential therapeutic targets, we interrogated these genes in the Drug-Gene Interaction Database (DGIdb)[26] which includes information from > 30 trusted sources on the druggable genome with potentially druggable genes (genes or gene products that are predicted to be druggable, e.g., kinases) and genes with evidence of drug-gene interaction (genes or gene products that have been tested and reported to be druggable). For each disease, the names of all of the likely causal genes were entered in the search box in the search engine interface (https://dgidb.org/), and all genes indicated to be druggable in DGIdb were recorded.

## Pooled CRISPR-Cas9-nuclease knockout screens

Pooled CRISPR-Cas9-nuclease knockout screens were performed on the genes that were identified in the colocalization analyses to be likely causal for CAD, hypertension, stroke, and/or AAA, and that were protein-coding genes because the pooled CRISPR-Cas9-nuclease screening technique is unable to screen non-coding RNA genes. A single-guide RNA (sgRNA) library was constructed, which included 4 sgRNAs for each of the target genes and 1000 non-targeting control sgRNAs. The sequences of the sgRNAs were selected from the Human CRISPR knockout Pooled Library (GeCKO v2 or Brunello). Corresponding oligonucleotides were synthesized by GenScript, and cloned into the lentiCRISPRv2 plasmid (Addgene, #52961) as described[89]. The quality of the library was analysed by next-generation sequencing and MAGeCK (Model-based Analysis of Genome-wide CRISPR-Cas9 Knockout)[90].

Human aortic smooth muscle cells (Lonza, CC-2571) were cultured in 10 plates (15 cm diameter) for the proliferation experiment and 5 plates for the migration experiment, with 3 technical replicates in each experiment. At 80% confluency, cells were transduced with the above-mentioned sgRNA library at a multiplicity of infection (MOI) of 0.2-0.3 as determined by titration. Two days post-transduction, cells were treated with 1 μg/mL puromycin and selected over the course of 8 days to enrich for transduced cells. After selection, the media was replaced with Human Vascular Smooth Muscle Cell Basal Medium (Thermo-Fisher Scientific, M231500) with Smooth Muscle Growth Supplement (ThermoFisher Scientific, S00725), 2% foetal bovine serum, and 1× penicillin-streptomycin.

For proliferation experiments, genomic DNA was prepared from 5 plates on day 5 and from another 5 plates on day 28, post-puromycin treatment. Genomic DNA was prepared using the PureLink Genomic DNA Mini Kit (ThermoFisher Scientific, K182002).

For migration experiments, on day 12 post-puromycin treatment, $8 \times 10^6$ cells in 1 mL Human Vascular Smooth Muscle Cell Basal Medium (ThermoFisher Scientific, M231500) without supplements, were seeded at $2.5 \times 10^6$ per 8 μm pore polycarbonate insert in 6-well (24 mm) transwell plates (Corning, 3428). The lower wells contained 2 mL Human Vascular Smooth Muscle Cell Basal Medium (ThermoFisher Scientific, M231500) with 100 ng/mL platelet-derived growth factor-BB (Sigma-Aldrich, P3201-50UG). After 24 h, cells from both the top and bottom surfaces of the insert were harvested separately with trypsinization and then subjected to genomic DNA extraction with the use of the PureLink Genomic DNA Mini Kit (ThermoFisher Scientific, K182002).

PCR with the use of KAPA HiFi HotStart Ready Mix (Roche, 07958927001) was performed on the genomic DNA samples (32 μg genomic DNA per sample) from the above-described proliferation and migration experiments. The sequences of the PCR primers were as described by Joung et al.[89] (Supplementary Data 33). The PCR cycling conditions were as follows: 95 °C for 5 min, then 24–26 cycles of 98 °C for 20 s, 60 °C for 15 s and 72 °C for 15 s, and finally 72 °C for 1 min.

The PCR products were pooled and purified using KAPA Hyper-Pure Beads (Roche, 08963843001) with a PCR product-to-bead ratio of 1:0.55. After collecting the supernatant an additional 0.35 volume of beads were added to achieve a final ratio of 1:0.9. Each purified sample was dissolved in 20 μL nuclease-free water, and the concentrations were quantified with the Qubit dsDNA HS Assay Kit (ThermoFisher Scientific, Q32851) and the quality assessed with DNA 1000 kit (Agilent, 5067-1504) on the 2100 Bioanalyzer System (Agilent). Subsequently, libraries with barcoded Rev-1 to Rev-4 were pooled (500 ng each sample, combined into a total volume of 40 μL) and further quantified by quantitative PCR using standards from the NEBNext Library Quant Kit for Illumina (New England Biolabs, E7630S). The combined libraries were sequenced on the Illumina NovaSeq X platform, at 6 million $2 \times 150$ bp paired-end reads per sample.

Raw reads files were processed with Cutadapt[91] for adaptor trimming with custom scripts. Subsequently, MAGeCK[90] (version 0.5.9.5) count and test algorithms were applied to the migration and proliferation datasets to identify positively-selected sgRNAs and negatively-selected sgRNAs, through paired analyses with control sgRNAs normalisation. For the proliferation experiment, data from day 28 were compared with data from day 5, post-puromycin treatment. For the migration experiment, data from the bottom surface of the insert were compared with data from the top surface of the insert. Data from the 3 replicates of the proliferation experiment had similar distributions and corrections, so did data from the 3 replicates of the migration experiment, indicating high reproductivity (Supplementary Fig. 13). Volcano plots were generated with the use of SRplot[81].

## Vascular smooth muscle cell proliferation assay

Human coronary artery SMCs (Cell Applications, 350K-05a) were transfected with either *CARF* siRNA (ThermoFisher Scientific, HSS128878) or control siRNA (ThermoFisher Scientific, 4390843), and then a proliferation assay was performed using Cell Counting Kit-8 (MedChemExpress, HY-K0301), with 4 technical replicates. In brief, cells were cultured in 96-well plates ($5 \times 10^4$ cells/well) for 48 h, and a 2-(2-methoxy-4-nitrophenyl)−3-(4-nitrophenyl)−5-(2,4disulfophenyl)−2H-tetrazolium solution was added into each well. At 2 h after adding the reagent, a colorimetric assay was carried out using a microplate reader which measured the absorbance at 450 nm.

Human aortic smooth muscle cells (Lonza, CC-2571) were transfected with either *BCAR1* DsiRNA (Integrated DNA Technologies, sequence described in Supplementary 33) or negative control DsiRNA (Integrated DNA Technologies, 51-01-14-04), with 3 technical replicates. At 0, 24, and 48 hours post-transfection, cells were subjected to a proliferation assay with the use of Cell Counting Kit-8 (Abcam, ab228554).

Human aortic smooth muscle cells (Lonza, CC-2571) were transfected with either *SMARCA4* siRNA (ThermoFisher Scientific, HSS110005) or non-targeting control siRNA (ThermoFisher Scientific, 12935300 l), with 8 technical replicates. At 0, 24, 48, 72, and 96 hours post-transfection, cells were subjected to a proliferation assay using Cell Counting Kit-8 (Abcam, ab228554). A serial dilution starting from 25,000 cells was generated to serve as a standard curve for analysing the results of the proliferation assay.

## Vascular smooth muscle cell migration assay

Human aortic smooth muscle cells (Lonza, CC-2571) were transfected with either *FES* siRNA (ThermoFisher Scientific, HSS103636), control siRNA (ThermoFisher Scientific, 12935300), *CSNK2B* DsiRNA (Integrated DNA Technologies, sequence described in Supplementary 33) or negative control DsiRNA (Integrated DNA Technologies, 51-01-14-04), with the use of Lipofectamine RNAiMAX Transfection Reagent (ThermoFisher Scientific, 13778150). At 47 hours post-transfection, cells were incubated with 1 μg/mL mitomycin C (Sigma-Aldrich, M7949-2MG) for 1 h and then $4 \times 10^4$ cells were seeded in each well of a Corning Transwell 24-well plate with 8 μm pores (Sigma-Aldrich, CLS3422-48EA), with 6 technical replicates per condition. 24 h later, cells were incubated with 4% formaldehyde (Sigma-Aldrich, F8775-500ML) for 10 min, 5% Triton X-100 (Sigma-Aldrich, T8787) for 5 minutes, and then 1 μg/mL DAPI (4′,6-diamidino-2-phenylindole) (ThermoFisher Scientific, 62248) for 10 minutes. Subsequently, 3 images of each well were taken using a fluorescence microscope with a 10 × objective lens and a digital camera. Thereafter, cells on the upper aspect of the transwell were wiped off, and 3 images of the lower aspect were taken. Cells in each image were counted with the use of ImageJ (version2.14.0) software, and the cell counts in each set of 3 images were averaged.

## RNA-seq

Human coronary artery smooth muscle cells (Cell Applications, 350K-05a) transfected with either *FES* siRNA (ThermoFisher Scientific, HSS103636) or control siRNA (ThermoFisher Scientific, 12935300), with 5 technical replicates per condition, were subjected to RNA-seq. A strand-specific library with rRNA removal was prepared from total RNA of each sample, and 150 bp paired-end sequencing at a > 30 million read depth (12 G raw data per sample) was performed with the Illumina NovaSeq 6000 Sequencing System. Raw data (raw reads) in FASTQ format were processed as follows: (1) removing reads with 5' adaptors, reads without 3' adaptor or insert sequence, reads with >10% undetermined nucleotides, reads with >50% nucleotides with Qphred ≤ 20, and reads with ploy A/T/G/C; (2) removing adaptor sequences from the 3' ends of reads. Thereafter, the processed clean reads from each sample were mapped to the human reference genome with HISAT2 software, and subsequently transcripts were assembled with the StringTie programme. For quantification of the transcripts and genes, RPKM (Reads Per Kilobase of transcript per Million mapped reads) values were determined with the use of StringTie. A differential expression analysis was performed with the use of Cuffdiff, and *P*-values were adjusted using the Benjamini and Hochberg method. A volcano plot of differentially-expressed genes (adjusted $P < 0.05$) was generated with the use of SPplot[81].

## Proteomics and phosphoproteomics analyses

Human coronary artery smooth muscle cells (Cell Applications, 350K-05a) transfected with either *FES* siRNA (ThermoFisher Scientific, HSS103636) or control siRNA (ThermoFisher Scientific, 12935300), with 5 technical replicates per condition, were subjected to quantitative proteomics and phosphoproteomics analysis. In brief, protein extracts from cells were reduced with dithiothreitol, alkylated with iodoacetamide, washed with acetone, and then dissolved in a solution containing 0.1 M triethylammonium bicarbonate (pH 8.5) and 6 M urea. The concentration of total protein in the samples was quantified using the Bradford method, and protein quality was assessed using sodium dodecyl sulfate-polyacrylamide gel electrophoresis with Coomassie Brilliant Blue R-250 staining. The protein samples were digested by incubation with trypsin and subjected to tandem mass tag (TMT) labelling using a mass tagging kit (ThermoFisher Scientific, 90113). Phosphopeptides in the samples were enriched using PHOS-Select Iron Affinity Gel (Sigma-Aldrich, P9740). Subsequently, liquid chromatography-mass spectrometry was performed with the use of an EASY-nLCTM 1200 System (ThermoFisher Scientific) and a Q Exactive HF-X mass spectrometer (ThermoFisher Scientific).

The liquid chromatography-mass spectrometry data were analysed using Proteome Discoverer Software 2.4 (ThermoFisher Scientific) with the following parameters: a mass tolerance of 10 ppm for precursor ion scans and a mass tolerance of 0.02 Da for the product ion scans. Phosphorylation of serine (S)/threonine (T)/tyrosine (Y), oxidation of methionine (M), and tandem mass tags (TMT)-plex of lysine (K) were specified as dynamic modifications. Carbamidomethyl was set as a fixed modification. TMT-plex, acetylation, and loss of methylation were specified as N8 terminal modifications. The motif-x algorithm was used to identify motifs enriched within a set of phosphosites. All enrichments were carried out for 7 residues surrounding the central residue with occurrences > 20 and $P < 10^{-6}$. WebLogo was used to build the motif figures. To predict kinase-substrate relationships, all identified serine, threonine and tyrosine phosphorylation sites (pS/T/Y) were scored with the NetPhorest algorithm.

Each protein was identified with at least 1 unique peptide at a false discovery rate (FDR) < 1%. Proteins with similar peptides that could not be distinguished based on the mass spectrometry data were combined as protein groups. Quantification based on intensity was performed.

A GO enrichment analysis was conducted on proteins that showed suggestive differences in modification between cells with siRNA-induced *FES* knockdown and cells transfected with the control siRNA ($P < 0.05$ by *t* test), using the default setting of Gene Ontology (http://geneontology.org). A bubble plot of enriched pathways was generated with the use of SPplot[81].

Any difference in the quantity of each protein between cells with FES knockdown and cells transfected with the control siRNA was ascertained by *t* test. A volcano plot of differentially expressed proteins (*P*-value < 0.05 and > 1.2fold) was generated with the use of SPplot[81].

## RT-PCR

Cells were lysed using TRIzol reagent (ThermoFisher Scientific, 15596026), and thereafter total RNA was extracted using the Direct-zol RNA MiniPrep kit (Zymo Research, R2050) according to the manufacturer's protocol. RNA was then reverse transcribed into cDNA with the use of iScript Reverse Transcription Supermix (Bio-Rad, 1708841). Subsequently, cDNA of the genes of interest was amplified using iTaq Universal SYBR Green Supermix (Bio-Rad, 1725124) and the primers described in Supplementary Data 35. The PCR thermocycling conditions were as follows: 95 °C for 2 minutes, followed by 40 cycles of 95 °C (denaturation) for 5 seconds and 60 °C (annealing and extension) for 30 s. Differences in expression levels of the genes of interest between cells transfected with the target gene siRNA and cells transfected with the control RNA were determined using the ΔΔCt method, with the housekeeping gene *ACTB* as a reference.

## Western blot analyses

Protein extracts from transfected cells were prepared using RIPA (radioimmunoprecipitation assay) lysis buffer (ThermoFisher Scientific, #89901), and protein concentrations in the extracts were quantified using Pierce BCA Protein Assay Kit (ThermoFisher Scientific, #23327). Equal amounts of protein from each sample were subjected to sodium dodecyl-sulfate polyacrylamide gel electrophoresis and thereafter transferred onto polyvinylidene difluoride membranes. The membranes were then blocked with 5% bovine serum albumin in Tris-buffered saline with Tween-20 for 2 h at room temperature, and subsequently incubated with a primary antibody (Supplementary Data 35) at 4 °C overnight. After washing, the membranes were incubated with a secondary antibody conjugated with horse-radish peroxidase (Supplementary Data 36) for 1 h at room temperature. The bands on the membrane were detected using a ChemiDoc Touch Imaging system (Bio-Rad).

## Animal study

The study was conducted in accordance with the UK Animals (Scientific Procedures) Act 1986, under a Home Office Project License (60/4332), and following the guidelines of ARRIVE (Animal Research: Reporting of In Vivo Experiments), the University of Leicester and the University of Oxford.

**Animals.** The generation of *Fes*[-/-]/*Apoe*[-/-] mice and *Fes*[+/+]/*Apoe*[-/-] control littermates has been described previously[28]. In brief, *Fes*[-/-] mice were backcrossed onto a C57BL6/J background for seven generations, and subsequently intercrossed with *Apoe*[-/-] mice (B6.129P2-Apoetm1Unc/J purchased from Jackson Laboratory) to generate *Fes*[-/-]/*Apoe*[-/-] mice and *Fes*[+/+]/*Apoe*[-/-] control littermates. Mice were housed in a pathogen-free environment. Animals were maintained in ventilated cages with a controlled 12:12-hour light/dark cycle (lights on at 07:00 h). The ambient room temperature was maintained at 22 ± 2 °C with a relative humidity of 55 ± 10%.

**Study of atherosclerosis.** *Fes*[-/-]/*Apoe*[-/-] and littermate control (*Fes*[+/+]/*Apoe*[-/-]) mice of 6 weeks of age fed a high fat diet for 12 weeks. The extent of aortic atherosclerotic lesions in the mice was analysed by *en face* staining of aortae with Oil Red O, as previously described[92]. In

brief, the aorta was opened longitudinally along the ventral midline from the iliac arteries to the aortic root, pinned out flat on a surface, and stained *en face* with Oil Red O. The stained aortas were photographed using a digital camera, and the Oil Red O-stained areas analysed by computer-assisted quantification using the Image-Pro software.

**Arterial pressure monitoring.** $Fes^{-/-}/Apoe^{-/-}$ and littermate control ($Fes^{+/+}/Apoe^{-/-}$) mice, at 8 weeks of age, were anaesthetised with 4% isoflurane in medical $O_2$ and maintained on a nose cone at 1.5-1% isoflurane for measurements of blood pressure and heart rate. The aorta was cannulated via the right common carotid artery using a 1.4 French blood pressure probe and transducer (Millar Instruments Inc. Houston, TX) coupled to a Powerlab/4SP data acquisition system (ADInstruments Pty Ltd., New Castle, Australia). Measurements were obtained after a 15 min equilibrium period; phenylephrine was dosed via a catheter placed in the jugular vein. Blood pressure and heart rate measurements were taken for 30 s and average recordings during this time taken. Increasing doses of phenylephrine (5 μg/kg and 25 μg/kg) were administered with a 5-minute wash out period between doses. After a further 10-minute wash out period, acetylcholine (5 μg/kg) was administered. Delta change in blood pressure was measured by subtracting baseline measurements from blood pressure after drug administration. A total of $n = 7$ $Fes^{+/+}/Apoe^{-/-}$ and 5 $Fes^{-/-}/Apoe^{-/-}$ mice per group were used in the study, with procedures performed blinded. One $Fes^{-/-}/Apoe^{-/-}$ mouse was excluded due to a technical issue with the catheterisation. Two hemodynamic measurements were excluded in the wild type group due to technical problems with the traces (one 5 μg/kg phenylephrine and one acetylcholine), and one wild type measurement at 5 μg/kg was missed due to a problem with the injection.

**Collapsing analyses of human FES protein truncating variants in relation to blood pressure and CAD traits**

Through the AstraZeneca PheWAS Portal[25], we collated human data of deleterious genetic variants in the protein-coding regions of the human *FES* gene in relation to blood pressure (data fields 4079 and 4080), hypertension (I10), and the CAD traits angina (I20) and myocardial infarction (I21), based on whole genome sequencing and phenotypic measurements of European participants in the UK Biobank. Collapsing analyses of the human FES protein truncating variants in relation to the blood pressure and CAD traits were performed.

**Statistical analyses**

The Mann-Whitney test or *t* test was used to ascertain differences between experimental groups in the VSMC proliferation assays, VSMC migration assays, RT-PCR analyses, standardised band intensity in the Western blot analyses, and atherosclerotic lesion areas and blood pressure in the animal studies, respectively.

**Reporting summary**

Further information on research design is available in the Nature Portfolio Reporting Summary linked to this article.

## Data availability

The data that support the findings of this study are available from the corresponding author. The RNA-Seq data and H3K27ac HiChIP-seq data are available from Gene Expression Omnibus with the accession numbers GSE189300 and GSE282557. The DNA methylation data are available from ArrayExpress with the accession ID E-MTAB-15426. The proteomics data and phosphoproteomics data are available from MassIVE with the accession IDs PXD061984 [https://massive.ucsd.edu/ProteoSAFe/private-dataset.jsp?task=8660bf01bde54a50899b9126643cf52c] and PXD061992 [https://massive.ucsd.edu/ProteoSAFe/private-dataset.jsp?task=

c1493e7d255a401b93782a307e5f0b64], respectively. Source data are provided in this paper.

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

## Acknowledgements

This work was funded by the British Heart Foundation [RG/16/13/32609, RG/19/9/34655, PG/16/9/31995, PG/18/73/34059, and SP/19/2/344612 to S.Y.), The National Medical Research Council of Singapore (MOH-001229 and MOH-001479 to S.Y.), and the National University of Singapore/National University Health System (NUHSRO/2022/004/Startup/01 to S.Y.). C.U.S. is a Leicester British Heart Foundation Accelerator Award (AA/18/3/34220) Research Fellow. D.G.M. is supported by the British Heart Foundation Research Excellence Award (RE/24/130031), the van Geest Foundation Heart and Cardiovascular Diseases Research Fund, and was awarded a BHF Accelerator Early Careers Researcher Interdisciplinary Fellowship and pump-priming funding from the Leicester British Heart Foundation Accelerator Award (AA/18/3/34220). This work falls under the portfolio of research conducted within the National Institute for Health Research Leicester Biomedical Research Centre and the Leicester BHF Centre of Research Excellence. This research used the ALICE High Performance Computing Facility at the University of Leicester.

## Author contributions

C.U.S., D.G.M., H.Z., and Y.Z. performed the analyses. D.G.M., C.A., P.G., L.T., D.S.S.S., H.Z., D.P.L., E.K., W.Y., J.C., R.C., K.E.H., C.G.A-N., H.L., M.J.D., P.Y.L, X.H., G.E.M., E.J.S., and G.D. performed the experiments. P.A.G. provided the mouse line. H.Y., R.S.Y.F., N.J.S., T.R.W., and S.Y. supervised the work. S.Y. wrote the paper. All the authors revised the manuscript, contributed with discussions and revisions and approved the final version of the manuscript.

## Competing interests

The authors declare no competing interests.

## Additional information

[1]Division of Cardiovascular Sciences, University of Leicester, Leicester, UK. [2]National Institute for Health Research Leicester Biomedical Research Centre, Leicester, UK. [3]Leicester British Heart Foundation Centre of Research Excellence, University of Leicester, Leicester, UK. [4]Cardiovascular-Metabolic Disease Translational Research Programme, Department of Medicine, National University of Singapore, Singapore, Singapore. [5]Division of Basic Medicine, Shantou University Medical College, Shantou, China. [6]Department of Pathophysiology, Gannan Medical University, Ganzhou, Jiangxi, China. [7]The First Affiliated Hospital of Shantou University Medical College, Shantou, China. [8]Montreal Heart Institute, Montreal, Canada. [9]Department of Pathology and Molecular Medicine, Queen's University, Kingston, Canada. [10]Division of Cardiovascular Medicine, Radcliffe Department of Medicine, University of Oxford, Oxford, UK. [11]Wellcome Trust Centre for Human Genetics, University of Oxford, Oxford, UK. [12]These authors contributed equally: Charles U. Solomon, David G. McVey. [13]These authors jointly supervised this work; Nilesh J. Samani, Tom R. Webb, Shu Ye. ✉e-mail: shuye68@nus.edu.sg

