## [Transparent Peer Review file · Nature Communications]

Integrative functional genomics analysis identifies pleiotropic genes for vascular diseases

Corresponding Author: Professor Shu Ye

Version 0:

Reviewer comments:

Reviewer #1

(Remarks to the Author)

Solomon et al. conducted a series of multi-step analyses to thoroughly examine how risk variants and their associated genes contribute to the development of vascular diseases, including coronary artery disease (CAD), hypertension, systolic blood pressure (SBP), diastolic blood pressure (DBP), and abdominal aortic aneurysm (AAA). Initially, the authors created a cohort of 1,846 samples in a previous study and performed genotyping alongside whole-genome RNA sequencing to produce eQTL datasets. They incorporated GWAS summary statistics for traits such as CAD, hypertension, SBP, DBP, and AAA to pinpoint causal loci and genes. Following this, the authors selected SNPs and analyzed the colocalization between eQTL data from smooth muscle cells and GWAS findings, identifying several pleiotropic genes associated with these traits. To validate their results, the authors conducted various analyses, including ATAC-seq, H3K27ac HiChIP-seq, and DNA methylation assessments. The authors proceeded with pathway enrichment analyses, identified harmful protein-coding variants, evaluated the druggability of likely causal genes, and performed CRISPR-Cas9 nuclease screening. Ultimately, the authors concentrated on the FES gene and carried out assays to evaluate smooth muscle cell proliferation and migration. These included RNA sequencing, proteomics, phosphoproteomics, RT-PCR, and western blot analyses to determine the function of FES, particularly focusing on the effects of its absence. Furthermore, knocking out the Fes gene in mice leads to increased atherosclerosis and elevated blood pressure.

The experimental design of this paper appears comprehensive and structured; however, the following comments and suggestions still need to be addressed:

Major comments:

Use of umbilical cord VSMCs for eQTL analysis:

It's unclear why umbilical cord-derived VSMCs were used for eQTL mapping in the context of adult cardiovascular disease. Neonatal cells differ substantially from adult cells in both gene expression and regulatory landscape. The authors should justify this choice and discuss its limitations.

Selection of GWAS SNPs based solely on genomic position (Line 339):

The strategy of keeping only the first SNP by genomic position within LD blocks ($R^2 > 0.8$) assumes that this SNP is the most representative, which is questionable. It would be helpful for the authors to clarify why this approach was used and discuss its limitations as well.

Colocalization strategy (Line 385):

Using a fixed 500kb window across all loci doesn't account for the differences in LD structure or regulatory architecture between regions. It's not clear why this specific window size was applied uniformly, as the window from locus to locus can vary. Also, the decision to retain only genes with more than 4 signals isn't explained, the authors should clarify the reasoning behind this threshold. This reviewer questions why the ones with less than 4 were not retained.

Deleterious protein-coding variants in likely causal genes (Lines 152-165):

This reviewer found this approach to be very interesting. Could the authors clarify why the same gene (supplementary table 18) exhibits both beneficial and detrimental effects concerning CAD or blood pressure for certain mutations?

Gene selection for validation (Lines 182-190):

The criteria for selecting CARF, BCAR1, and FES for experimental validation are not explained. In addition, how did the focus suddenly shift to just FES. If the others did not reach significance, this should be mentioned and the negative results made available in supplementary materials.

Fes knockout model and sex bias in in-vivo experiments:

There's a disconnect between the in vitro and in vivo data for FES. Knockout in VSMCs improved proliferation and

migration, which can be beneficial in vascular repair and fibrous cap stability, yet the whole-body knockout in mice led to increased atherosclerosis. This contradiction isn't addressed. Additionally, FES is broadly expressed across multiple cell types; the plaque phenotype likely reflects cumulative effects beyond VSMCs. To resolve this, a VSMC-specific knockout will clarify if the effect is truly VSMC-driven. Also, only male mice were used. If female mice were tested and showed no effect, that should be reported. If not tested at all, that's a major limitation; sex differences in vascular disease are well established and should be considered. Lastly, it has been shown that FES has a sex-biased effect on Blood pressure (PMID: 38459180).

Minor comments:

Lack of detail in key results: Several important findings are mentioned only briefly without sufficient interpretation. The results section can benefit from more throughout reporting.

Figure referencing issues: There are several incorrect figure references in the results section (eg. Lines 193, 200, 203, 205, 217, 223, 229, 231). These should be checked and corrected.

Supplementary Figure 10: P-value is missing.

Line 38: The introduction of FES is too abrupt, a brief transition of context will help improve the flow. Also, the sentence "FES is a pleotropic for both ..." is incomplete, pleiotropic what? Please clarify.

Line 202: Please explain briefly the relevance of the two genes MMP1 and MMP3.

Line 202: The full name "epidermal growth factor" should be written out before introducing the abbreviations (EGF), not the opposite.

Reviewer #2

(Remarks to the Author)

Solomon and colleagues presented their work characterizing the causative genes associated with vascular disease. They applied multi-omics experiments, including ATAC-seq, H3K27ac HiChIP-seq, and DNA methylation profiling, to explore the functional genetic variants identified in their previous eQTL/GWAS study. They identified 23 genes that were commonly significant for more than one vascular disease. These genes were further examined using the UK Biobank and Finnish Biobank datasets. The authors also utilized CRISPR screens to identify important genes associated with CSMC migration or proliferation, selecting FES for further functional validation. The study demonstrated the function of FES in human vascular disease through the integration of distinct omics data, providing a valuable strategy and resources to the scientific community. However, the manuscript overall lacks detail, and most of the analyses do not seem to have been comprehensively performed. Here are my concerns.

1. The manuscript lacks clarity on how many samples or replicates were performed for each omics experiment. The bioinformatics analysis is not sufficiently detailed, providing only conclusions without describing the methodologies and data quality. It is also noted that the data is not open-access. For example, the authors mention that many variants are located in open chromatin regions (OCRs) with low methylation, but how are these lowly methylated regions defined? What criteria and number of replicates were used? Were any differential methylation or chromatin accessibility analyses performed? Similar issues are present in the HiChIP-seq data analysis. How does H3K27ac correlate with ATAC-seq and DNA methylation at significant genetic variants?

2. While the approaches were used to identify causal genes, the identified genes do not seem significant in the eQTL/GWAS analysis. This might suggest that the regulation of these causal genes is more critical to human vascular disease. However, the current results do not explore the regulatory relationships between genetic variants and their target genes. Additionally, it is unclear how these regulatory relationships are distributed within the human population to connect with vascular disease.

3. The CRISPR screening identified important gene candidates, but the log fold change appears relatively small in this analysis. It is also unclear whether the y-axis represents p-value or adjusted p-value, and what the stringency of this analysis was. In Figure 5B, the blue and pink dots are marked as up in non-migrated cells, but further clarification is needed.

4. The animal experiments indicate that downregulated expression of FES is associated with increased atherosclerotic lesions and hypertension. Are these results consistent with human data?

Version 1:

Reviewer comments:

Reviewer #1

(Remarks to the Author)

Colocalization strategy

For eCAVIAR and SMR/HEIDI analyses, can the author create a locus zoom window using the GWAS Tag SNP by building 200 KB (as a starting point) around significant SNPs based on LD, then adding 100 or 200 kb left or right until no SNP is in LD?

Fes knockout model and sex bias in in-vivo experiments:

Since in vitro FES gene analyses were conducted in VSMCs, unlike the in vivo mouse model with a whole-body knockout, it is difficult to determine the cell type source of the effect, making it hard to draw conclusive results. Therefore, it would be beneficial to have VSMC-specific KO of Fes and study atherosclerosis.

Additionally, combining data from both males and females with a small sample size, especially in females, for the atherosclerosis study, is a concern and makes it hard to identify if there is a sex-specific effect.

Reviewer #2

(Remarks to the Author)

I appreciate that the authors addressed all my concerns.

Reviewer #3

(Remarks to the Author)

Report

Report focusing on the Fes knockout mouse experiments and the in-vitro validation data:

The group has previously generated an eQTL catalogue for human umbilical cord smooth muscle cells (SMC) and found co-localisation of SMC eQTLs with coronary artery disease (CAD)-associated risk loci. Here, they report overlap of hypertension, stroke and abdominal aortic aneurysm risk variants with SMC eQTLs and identify variant shared for multiple conditions. Co-localisation analysis and epigenetic profiling data is leveraged to identify candidate causal genes and associated mechanisms. The analysis suggests that some genes are linked to multiple conditions suggesting pleiotropic effects or common disease pathways for the conditions. Furthermore, database analysis highlights many of the identified genes as potential drug targets.

Pooled CRISPR screens of genes regulated by eQTLs colocalising with risk variants was done in human aortic smooth muscle cells to reveal impact on proliferation (enrichment at day 28 vs d 5) and migration (transwell migration). Validation experiments confirmed changes for 4 out of 5 genes selected based on suggested to have pleiotropic effects and top ranked in the CRISPR screen.

- Significance (grey dots in Figure 5A) in the CRISPR screen appears to be evaluated without adjusting for multiple testing, which is problematic. The screen was done in triplicate, but it is not clear to me whether cells from different donors were used for the three screen replicates and how the replicates were considered for ranking of genes.
- For the CRISPR screen validation experiments, it is unclear how experimental replication is achieved; the legend to figure 5 states group sizes of 4 or 6, but methods section suggest that these are technical repeats. For experiments shown in supplementary data, there are varying group sizes and timings (e.g. SMARCA evaluated at up to 96h). Ideally cells from different donors should be used for each experiment for datapoints to be considered independent.

FES, which showed an impact on migration and genetic association with CAD, was selected for further analysis. The group has previously shown increased atherosclerosis in Fes knockout (KO) animals (PMID: 36321446) and this was here confirmed used an alternative measurement in the same experimental atherosclerosis model. Blood pressure measurements in non-atherogenic mice suggest that Fes-KO impacts both base line pressures and the response to acetylcholine. Furthermore, genetic variants resulting in truncated FES protein was associated with elevated blood pressure and likelihood of hypertension, angina and myocardial infarction in human.

- The group size for blood pressure analysis is low, with only 4 animals in the KO group. This is a concern as the pressure is measured under anaesthesia (isoflurane, which can cause peripheral vasoconstriction), which is more variable. The low animal numbers also preclude comparison of effects in male and females.
- The analysis of human data is not stratified by sex or analysed for confounding factors

Molecular changes of siRNA-mediated FES knockdown in cultured SMCs are recorded by RNA-seq and proteomics analysis. This reveals changes in pathways and specific proteins, including MMP1.

- The proteomics and phospho-proteomics only provide uncorrected p-values, with little or no, experimental validation. It is not clear whether and how these changes are related to the in vivo phenotypes in mouse (or human traits), nor is the phenotype directly linked to an effect of FES on SMCs.

Version 2:

Reviewer comments:

Reviewer #1

(Remarks to the Author)

The authors addressed all my concerns.

Reviewer #3

(Remarks to the Author)

The revised manuscript have responded to most of the queries raised in my report, but the critique regarding group sizes for the in vivo analysis is not addressed (listed below). The new information about replication and the appropriate statistical analysis raise some concerns regarding the robustness and reproducibility about the findings from the experimental analysis. The revised manuscript state that the study does not aim to test whether the effects in VSMCs account for detected risk heritability, but to add a mechanistic hypothesis.

Additions to the revised manuscript

1. Appropriate statistical analysis (adjusted p-values) has been included for the crisp screen. Adjusted p-values suggest that most observed changes are unlikely to be due to actual biological differences. While FES scores 2nd in the migration screen, the adjusted p-value is 0.92
2. The confusion regarding the nature of the replication has been resolved. This reveals that all CRISPR screen replicates and the follow up experiments were done in cells from a single donor. These two sets of experiments were done using different loss-of-function technologies. However, using a single genetic background means that the robustness of the observation is not tested.
3. The authors clarify that comparison between males and females is not intended. However, this does not address the concern regarding small group sizes
4. Concern regarding confounding factors has been appropriately addressed
5. Appropriate statistical analysis (adjusted p-values) has been included for the proteomics analysis. This reveals that most observed changes are unlikely to be due to actual biological differences, which does not justify the statement that "The phosphoproteomics analysis detected many differentially phosphorylated proteins (Supplementary Table 26)...". However, the text has been revised to tone down the conclusions based on this data.
6. The discussion states that the manuscript does not intend to assess cell-type specificity.

Version 3:

Reviewer comments:

Reviewer #3

(Remarks to the Author)

I have no further comments

Response to Reviewers' Comments

We thank the reviewers for their highly valuable comments and constructive advice. In the revised manuscript, the reviewers' points have been addressed as detailed below. Reviewers' comments are in bold font, followed by authors' response in regular font.

Reviewer 1

Solomon et al. conducted a series of multi-step analyses to thoroughly examine how risk variants and their associated genes contribute to the development of vascular diseases, including coronary artery disease (CAD), hypertension, systolic blood pressure (SBP), diastolic blood pressure (DBP), and abdominal aortic aneurysm (AAA). Initially, the authors created a cohort of 1,846 samples in a previous study and performed genotyping alongside whole-genome RNA sequencing to produce eQTL datasets. They incorporated GWAS summary statistics for traits such as CAD, hypertension, SBP, DBP, and AAA to pinpoint causal loci and genes. Following this, the authors selected SNPs and analyzed the colocalization between eQTL data from smooth muscle cells and GWAS findings, identifying several pleiotropic genes associated with these traits. To validate their results, the authors conducted various analyses, including ATAC-seq, H3K27ac HiChIP-seq, and DNA methylation assessments. The authors proceeded with pathway enrichment analyses, identified harmful protein-coding variants, evaluated the druggability of likely causal genes, and performed CRISPR-Cas9 nuclease screening. Ultimately, the authors concentrated on the FES gene and carried out assays to evaluate smooth muscle cell proliferation and migration. These included RNA sequencing, proteomics, phosphoproteomics, RT-PCR, and western blot analyses to determine the function of FES, particularly focusing on the effects of its absence. Furthermore, knocking out the Fes gene in mice leads to increased atherosclerosis and elevated blood pressure. The experimental design of this paper appears comprehensive and structured; however, the following comments and suggestions still need to be addressed:

Major comments:

Use of umbilical cord VSMCs for eQTL analysis:

It's unclear why umbilical cord-derived VSMCs were used for eQTL mapping in the context of adult cardiovascular disease. Neonatal cells differ substantially from adult cells in both gene expression and regulatory landscape. The authors should justify this choice and discuss its limitations.

We thank the reviewer for the constructive comment and advice. We have added the following text in the Discussion:

"We note some limitations of our work. First, our study used human umbilical cord-derived VSMCs for eQTL mapping in the context of adult cardiovascular disease. At present, a large collection of adult VSMCs is unavailable. Therefore, in this study we utilized a large collection of human umbilical cord-derived VSMCs (VSMC eQTL data from 1,486 individuals) to provide high statistical power for the comprehensive genome-wide eQTL analysis. As the cells were isolated from umbilical cords from individuals of the same biological age, the analysis also benefited from circumventing age-related confounding. However, there is a caveat that neonatal cells can differ from adult cells in gene regulation and expression. Notwithstanding, in a recent work¹⁶, we compared the transcriptomes of the umbilical cord-derived VSMCs utilized in this study with reported transcriptomic data from other types of cell/tissue^{58, 59}, and found that the gene expression profile of the human umbilical cord-derived VSMCs mapped closer to adult human coronary artery smooth muscle cells than any other cell/tissue types¹⁶. ..." (page 10/paragraph 3/line 1-12).

Selection of GWAS SNPs based solely on genomic position (Line 339):

The strategy of keeping only the first SNP by genomic position within LD blocks ($R^2 > 0.8$) assumes

that this SNP is the most representative, which is questionable. It would be helpful for the authors to clarify why this approach was used and discuss its limitations as well.

We apologize for the lack of clarity in our original manuscript regarding the specific purpose that these SNPs were selected for. These SNPs were selected to tag the genetic variants/haplotypes which had been reported to be associated with CAD, blood pressure, hypertension, stroke, and AAA, respectively, in GWAS. These tagging SNPs were then looked up in the VSMC eQTL catalogue generated in our study, so as to investigate which of the tagging SNPs (or SNPs in LD with these tagging SNPs) showed eQTL effect in the VSMC eQTL catalogue. This part of the study was not intended to identify the causal variants in the disease-associated LD blocks. Therefore, this was followed by the colocalization analyses to identify likely causal variants. The colocalization analyses (eCAVIAR, and SMR/HEIDI) included all SNPs that have been reported to be associated with the relevant disease (CAD, hypertension, stroke, and AAA, respectively) in GWAS, rather than just the above selected tagging SNPs.

In the revised manuscript, we have tried to clarify this by revising the relevant texts to read as follows:

“We recently established a large collection of VSMCs from different individuals (n = 1,486), and on this cell bank we performed genome-wide genotyping and whole-genome RNA-sequencing (RNA-seq) to generate a comprehensive catalogue of expression quantitative trait loci (eQTL) with stringent statistical significance¹⁶. We compiled a list of tagging single nucleotide polymorphisms (SNPs) that marked the genetic variants/haplotypes reported to be associated with CAD in GWAS^{1, 2, 3, 4} and then interrogated these tagging SNPs in the VSMC eQTL catalogue. As reported previously¹⁶, we observed that ~60% of the CAD-associated SNPs had an eQTL effect on gene expression, with two-thirds of them affecting more than one gene” (page 4/paragraph 1).

“Here we compiled lists of tagging SNPs that marked the genetic variants/haplotypes reported to be associated with blood pressure/hypertension^{5, 19}, stroke^{5, 6, 7} and/or AAA^{5, 8} in GWAS, and interrogated these tagging SNPs in the abovementioned VSMC eQTL catalogue. We found that ~59% of the blood pressure/hypertension-associated SNPs, ~55% of the stroke-associated SNPs, and ~59% of the AAA-associated SNPs, had eQTL effects in VSMCs (Supplementary Table 1-3 and Supplementary Fig. 1). Of these SNPs, ~66% showed eQTL effects on more than one gene. Of the associated genes, ~77% were protein-coding and ~15% were long non-coding RNAs (lncRNAs) (highlighted in green and yellow, respectively, in Supplementary Table 1-3).” (page 4/paragraph 2/line 1-8).

“To explore causal variants and related causal genes for each of the vascular diseases mentioned above, we performed colocalization tests, including eCAVIAR (eQTL and GWAS Causal Variant Identification in Associated Regions)²⁰ and SMR/HEIDI (summary-data-based Mendelian Randomization/Heterogeneity in Dependent Instruments)²¹, utilizing reported disease GWAS summary statistics in conjunction with the above-described VSMC eQTL dataset. These analyses detected panels of likely causal variants and likely causal genes for each of these vascular diseases. Supplementary Tables 5-8 show the results of these colocalization analyses, including the identified likely causal variants (columns A-F), the identified likely causal genes together with eQTL data in relation to the likely causal variants (columns G-M), summary statistics of reported associations of the likely causal variants with the relevant vascular diseases (columns N-R), and the statistics of the eCAVIAR and SMR/HEIDI analyses (columns S-Y).” (page 4/paragraph 3).

“Compilation of tagging SNPs that marked the genetic variants/haplotypes reported to be associated with CAD, blood pressure/hypertension, stroke, and AAA, respectively

We previously identified 424 tagging SNPs which marked the genetic variants/haplotypes that had been reported to be associated with CAD at the genome-wide significance level ($P < 5 \times 10^{-8}$) in GWAS^{1, 3, 4, 62, 63}, did not have significant linkage disequilibrium (LD) between them, and were typed in

the abovementioned VSMC collection either directly by using the Global Screening Array or indirectly by imputation¹⁶. In the present study, we further compiled lists of tagging SNPs which marked the genetic variants/haplotypes that had been reported to be associated with blood pressure/hypertension^{5, 19}, stroke^{5, 6, 7}, and AAA^{5, 8}, respectively, at the genome-wide significance level ($P < 5 \times 10^{-8}$) in GWAS, did not have significant LD between them, and were typed in the abovementioned VSMC collection either directly by using the Global Screening Array or indirectly by imputation¹⁶. LD between the SNPs was assessed using LDlink's SNPclip program, which removed SNPs based on LD in a given population and kept the first SNP (based on genomic position). We used an R^2 threshold of 0.8 and a MAF threshold of 0 with the "European" option. If there was available LD data, SNPs were included/excluded as required but if not, the SNPs were kept in the list and assumed to be independent signals. In total, 1402 independent SNPs were identified for blood pressure/hypertension (Supplementary Table 29), 139 for stroke (Supplementary Table 30), and 157 for AAA (Supplementary Table 31). Of these, 1203 for blood pressure/hypertension, 119 for stroke, and 136 for AAA, were available in this study (either directly or via proxy SNPs with $R^2 > 0.8$), which were typed in the abovementioned VSMC collection¹⁶ by the Global Screening Array or imputed, and were not filtered out (missingness < 0.05 , MAF > 0.01 ; HWE P-value $> 1 \times 10^{-6}$). (page 11/paragraph 3).

Colocalization strategy (Line 385):

Using a fixed 500kb window across all loci doesn't account for the differences in LD structure or regulatory architecture between regions. It's not clear why this specific window size was applied uniformly, as the window from locus to locus can vary.

Thank you for pointing this out. We have added the following text in the revised manuscript to describe the reason and the limitation:

"We analyzed 500kb windows as in several other studies^{16, 58, 65}. 500kb windows were more extensive than 200kb windows used in the originally reported eCAVIAR method²⁰; nevertheless, fixed windows of any size have the limitation of not accounting for between-region differences in LD structure." (page 12/paragraph 3/line 5-8).

Also, the decision to retain only genes with more than 4 signals isn't explained, the authors should clarify the reasoning behind this threshold. This reviewer questions why the ones with less than 4 were not retained.

We have realized that the original text "The eCAVIAR colocalization test was performed for genes that had > 4 overlapping SNPs with GWAS summary data" is unclear and misleading. We have amended it to read as follows:

"The eCAVIAR colocalization test was performed for loci that had 5 or > 5 overlapping SNPs present in both the eQTL and GWAS datasets, to reduce potential false positives due to overlaps by chance." (page 12/paragraph 3/line 8-10).

Due to the very large sizes of the eQTL and GWAS datasets each containing multiple millions of SNPs, the eCAVIAR analysis to look for overlaps of SNPs between the two datasets can detect overlaps of SNPs by chance. In a preliminary analysis, we observed that at loci where there was only 1 overlapping SNP when comparing the two datasets, the overlapping SNP was considered to be significant [colocalization posterior probability (CLPP) > 0.05 in the eCAVIAR analysis] at all (100%) of these loci and at least one eQTL gene (associated with this overlapping SNP) was considered to be significant (CLPP > 0.05) at nearly all (95%) of these loci (please refer to the plots below). It is unlikely that these are all true positive findings but likely that a proportion of them are false positives. However, if the analysis focused on loci where there were 5 or > 5 overlapping SNPs between the two datasets, only $\sim 10\%$ of the eQTL SNPs and $\sim 10\%$ of the eQTL genes were considered to be significant (CLPP > 0.05) in the eCAVIAR analysis (please refer to the plots below). Therefore, we

focused the eCAVIAR analysis of our study on loci that had 5 or > 5 overlapping SNPs present in both the eQTL and GWAS datasets, assuming that ~10% of eQTL SNPs are causal as such assumption (10%) has been applied in other studies [PMID: 29942083]. The higher likelihood of detecting potential false positives at loci where there are less than 5 overlapping SNPs in colocalization analyses have been noted by other investigators (<https://platform-docs.opentargets.org/gentropy/colocalisation>).

Deleterious protein-coding variants in likely causal genes (Lines 152-165):

This reviewer found this approach to be very interesting. Could the authors clarify why the same gene (supplementary table 18) exhibits both beneficial and detrimental effects concerning CAD or blood pressure for certain mutations?

We have realized that the word “Benign” in green color in the “Predicted consequences” column of this table (Supplementary Table 18 in the original manuscript; Supplementary Table 15 in the revised manuscript) may mislead and be interpreted as being beneficial, leading to the impression that some of the genes have both beneficial and detrimental mutations, although that is not the case. Therefore, we have changed the word “Benign” to “Non-deleterious missense variant” in this table, with the aim to make it clearer that the variant is predicted not to cause a deleterious effect on the function of the protein encoded by the gene.

Gene selection for validation (Lines 182–190):

The criteria for selecting CARF, BCAR1, and FES for experimental validation are not explained. In

addition, how did the focus suddenly shift to just FES. If the others did not reach significance, this should be mentioned and the negative results made available in supplementary materials.

Many thanks for the constructive comment and advice. We have revised the text accordingly to read as follows and added the negative result in Supplemental Fig. 10C.

“To assess possible effects of the likely causal genes on VSMC behavior, we performed high-throughput pooled CRISPR-Cas9-nuclease knockout screens targeting the protein-coding genes among these likely causal genes (this technique is unable to test non-coding RNA genes). The screens identified multiple genes that affected VSMC proliferation or migratory ability (Supplementary Table 24&25, and Fig. 5A&B). A selection of these genes was subjected to validation experiments, with the following selection criteria: 1) among the top 10 genes ranked by score in the pooled CRISPR-Cas9-nuclease knockout screens; 2) considered to be a pleiotropic gene for more than one vascular disease in the aforementioned cross-trait analysis; and 3) likely to be druggable in the abovementioned druggability analysis. In the validation experiments, VSMCs were transfected with either siRNA for the selected gene or negative control siRNA, followed by VSMC proliferation or migration assay. The following genes met the selection criteria and were tested in the validation experiments: *BCAR1*, *CARF*, and *SMARCA4* for VSMC proliferation, and *CSNK2B* and *FES* for VSMC migration. The validation experiments confirmed the CRISPR-Cas9-nuclease knockout screen findings for *BCAR1*, *CARF*, *FES*, and *SMARCA4*, with the same direction of effect in both the screen and validation experiments (Fig. 5C&D, and Supplementary Fig. 9A&9B), but not for *CSNK2B* (Supplementary Fig. 9C).” (page 6/paragraph 3/line 1-16).

Fes knockout model and sex bias in in-vivo experiments:

There’s a disconnect between the in vitro and in vivo data for FES. Knockout in VSMCs improved proliferation and migration, which can be beneficial in vascular repair and fibrous cap stability, yet the whole-body knockout in mice led to increased atherosclerosis. This contradiction isn’t addressed.

We appreciate the reviewer’s comment regarding the divergent roles of VSMCs in atherosclerosis, and have added the follow text in the Discussion:

“It has long been documented and is recently re-emphasized^{50, 51} that VSMC dedifferentiation and their accumulation significantly contribute to atherosclerotic plaque formation and enlargement, although VSMCs in the fibrotic cap are beneficial for plaque stability.” (page 10/paragraph 1/line 2-4).

Additionally, FES is broadly expressed across multiple cell types; the plaque phenotype likely reflects cumulative effects beyond VSMCs. To resolve this, a VSMC-specific knockout will clarify if the effect is truly VSMC-driven.

We appreciate the reviewer’s comment and have added the follow text in the limitations paragraph in the Discussion:

“... as *FES* is expressed in not only VSMCs but also some other cell types such as monocytes and endothelial cells, the total atherogenic contribution of *FES* is expected to derive from its effect on the various cell types, including VSMCs as suggested here and monocytes as indicated in our recent study²⁸. The relative contributions from the different cell types are yet to be established.” (page 10/paragraph 3/line 12-16).

Also, only male mice were used. If female mice were tested and showed no effect, that should be reported. If not tested at all, that’s a major limitation; sex differences in vascular disease are well established and should be considered. Lastly, it has been shown that FES has a sex-biased effect on Blood pressure (PMID: 38459180).

Thank you for the constructive comment and advice. We have added/annotated data from female mice on atherosclerosis and blood pressure in Figure 8 (8A, 8B & 8C) and Supplementary Figure 10 (10A & 10B).

Minor comments:

Lack of detail in key results: Several important findings are mentioned only briefly without sufficient interpretation. The results section can benefit from more throughout reporting.

Thank you for your constructive comment and advice. We have substantially revised the text of the Results section, with the intention to elaborate the key findings with more details and interpretation.

Figure referencing issues: There are several incorrect figure references in the results section (eg. Lines 193, 200, 203, 205, 217, 223, 229, 231). These should be checked and corrected.

Many thanks for pointing these out. We have checked and made corrections accordingly.

Supplementary Figure 10: P-value is missing.

Many thanks. P-values have been added to the Figure (This Supplementary Figure is labeled Supplementary Figure 9 in the revised manuscript).

Line 38: The introduction of FES is too abrupt, a brief transition of context will help improve the flow. Also, the sentence “FES is a pleiotropic for both ...” is incomplete, pleiotropic what? Please clarify.

Thank you for your constructive comment and valuable advice. As advised, we have revised the text to read as follows:

“Pooled CRISPR screen analyses of these likely causal genes indicate that many of them influence vascular smooth muscle cell behavior, and validation experiments of selected genes confirm that *FES*, *BCAR1*, *CARF* and *SMARCA4* exert such effects. Further functional experiments focusing on *FES*, a pleiotropic gene for both coronary artery disease and hypertension, show that ...” (Abstract, line 8-11).

Line 202: Please explain briefly the relevance of the two genes MMP1 and MMP3.

We have revised the text to read as follows:

“..., including increased expression of the extracellular matrix protein-degrading enzymes matrix metalloproteinase-1 (MMP1) and MMP3 (Fig. 6B&C, Supplementary Table 27&28).” (page 7/paragraph 1/line 9-10).

Additionally, the Discussion includes the following text:

“... both MMP1 and MMP3 are capable of degrading a variety of extracellular proteins in the vascular wall and have been shown to facilitate VSMC migration, matrix degradation and vascular remodeling⁵². These processes are instrumental in the pathogenesis of both atherosclerosis and hypertension^{52, 53, 54}.” (page 10/paragraph 1/line 5-8)

Line 202: The full name “epidermal growth factor” should be written out before introducing the abbreviations (EGF), not the opposite.

Thank you. As advised, we have changed the text to “epidermal growth factor (EGF)” (page 7/paragraph 1/line 6).

Reviewer 2

Solomon and colleagues presented their work characterizing the causative genes associated with vascular disease. They applied multi-omics experiments, including ATAC-seq, H3K27ac HiChIP-seq, and DNA methylation profiling, to explore the functional genetic variants identified in their previous eQTL/GWAS study. They identified 23 genes that were commonly significant for more than one vascular disease. These genes were further examined using the UK Biobank and Finnish Biobank datasets. The authors also utilized CRISPR screens to identify important genes associated with CSMC migration or proliferation, selecting FES for further functional validation. The study demonstrated the function of FES in human vascular disease through the integration of distinct omics data, providing a valuable strategy and resources to the scientific community. However, the manuscript overall lacks detail, and most of the analyses do not seem to have been comprehensively performed. Here are my concerns.

1. The manuscript lacks clarity on how many samples or replicates were performed for each omics experiment.

We thank the reviewer for the constructive comment. We have added such information in the Methods section (page 13/paragraph 2/line 1, page 13/paragraph 2/line 1-3, page 14/paragraph 2/line 1, page 17/paragraph 5/line 3, page 18/paragraph 2/line 3).

The bioinformatics analysis is not sufficiently detailed, providing only conclusions without describing the methodologies and data quality.

Thank you for the constructive comment. We have added relevant details in the Methods section (page 14/paragraph 4, page 14/paragraph 6, page 15/paragraph 2, page 15/paragraph 3).

It is also noted that the data is not open-access.

The data that support the findings of this study are available from the corresponding author upon reasonable request. Source data are provided with this paper. The RNA-Seq data and H3K27ac HiChIP-seq data are available from Gene Expression Omnibus with the accession number GSE189300 and GSE282557. The DNA methylation data are available from ArrayExpress with the accession ID E-MTAB-15426. The proteomics data and phosphoproteomics data are available from MassIVE with the accession IDs PXD061984 and PXD061992, respectively. (page 21).

We are not permitted to deposit the ATAC-seq data as they contain genetic sequences of donors of human umbilical cords, that are ethically considered sensitive personal data.

For example, the authors mention that many variants are located in open chromatin regions (OCRs) with low methylation, but how are these lowly methylated regions defined? What criteria and number of replicates were used?

Thank you for your constructive comment. We have expanded the DNA methylation analysis method section to include information regarding replicates and how low methylation was defined, as follows:

“DNA methylation assay was performed on 3 biological replicates of human umbilical artery smooth muscle cells (HUASMC, from different individuals) and on human aortic smooth muscle cells (HAoSMC, from 1 individual). ...” (page 14/paragraph 2/line 1-3).

“The hg38 genomic positions of the CpGs in the Infinium MethylationEPIC array were identified and the mean methylation beta value (taken from 3 HUASMC and 1 HAoSMC cell lines) was calculated for each CpG. A CpG was defined as having low methylation levels if the mean beta value was ≤ 0.25 . The distance between each CpG and the likely causal variant identified from our colocalization

analyses was calculated, and CpG-variant pairs with an absolute distance of ≤ 200 bp were considered to be within close proximity of each other (for variants with more than one CpG within close proximity, the nearest CpG was selected). Of the close-proximity CpG-variant pairs identified, the proportion that had low methylation (defined above) was calculated (both separately for each vascular disease and combined).” (page 14/paragraph 3).

Were any differential methylation or chromatin accessibility analyses performed?

The methylation and chromatin accessibility assays were performed on VSMCs from several individuals, which were intended to identify regions of low methylation and high chromatin accessibility, but were not powered for differential analyses. Therefore, differential analyses were not performed.

Similar issues are present in the HiChIP-seq data analysis. How does H3K27ac correlate with ATACseq and DNA methylation at significant genetic variants?

Thank you for your valuable comment. We have now placed the results of H3K27ac, ATAC-seq and DNA methylation, side by side, for each of the significant genetic variants in Supplementary Table 12, which shows that a large proportion of the significant genetic variants are located in regions with H3K27ac as well as ATAC peaks and/or low DNA methylation.

Additionally, we have added the following text in the Results section:

“To investigate if any of the likely causal variants identified in the above colocalization analyses were located in transcriptionally active regions of the genome, we performed VSMC ATAC-seq (assay for transposase-accessible chromatin with sequencing) to delineate open chromatin regions, as well as DNA methylation assay and H3K27ac HiChIP-seq (histone H3 acetylation at lysine 27 high throughput chromosome conformation capture with chromatin immunoprecipitation with sequencing) since DNA demethylation and H3K27 acetylation upregulate gene expression. We interrogated the likely causal variants in regions of ATAC-seq peaks, low DNA methylation and H3K27ac, respectively. The analyses showed that a large proportion of the likely causal variants were located in regions of ATAC-seq open chromatin (Supplementary Table 9), low DNA-methylation (variants highlighted in yellow in Supplementary Table 10), and/or H3K27ac (Supplementary Table 11). The likely causal variants in relation to peaks of ATAC-seq, low DNA methylation and H3K27ac, as well as their associated eQTL genes and vascular diseases, are summarized in Supplementary Table 12. As an example, we found that the CAD associated variant rs1894401 in the *FES* gene resided in an ATAC-seq open chromatin region with lessened DNA-methylation (Supplementary Fig. 2&3) and with H3K27ac (Supplementary Fig. 4).” (page 4/paragraph 4).

2. While the approaches were used to identify causal genes, the identified genes do not seem significant in the eQTL/GWAS analysis. This might suggest that the regulation of these causal genes is more critical to human vascular disease. However, the current results do not explore the regulatory relationships between genetic variants and their target genes. Additionally, it is unclear how these regulatory relationships are distributed within the human population to connect with vascular disease.

Many thanks for your valuable and constructive comment. We have now removed some of the variants, as they are not considered to be significant at the genome-wide significance threshold. In the revised manuscript, all the likely causal variants and their associated likely causal genes shown are significant in both GWAS and eQTL.

To show the relationship of the genetic variants with their target genes and human vascular disease, we have re-arranged the data in the relevant tables as follows:

- In Supplementary Tables 5-8, the names of the causal SNPs, their GWAS summary statistics (including P-values), the names of the causal genes, and their eQTLs summary statistics (including P-values), are shown one after the other.
- In Supplementary Table 12, information of regulatory variants, ATAC-seq peaks, H3K27ac peaks, low DNA methylation regions, target gene eQTL summary statistics and disease GWAS summary statistics, are placed one after the other.

Additionally, as mentioned above, we have added the following text in the Results:

“To investigate if any of the likely causal variants identified in the above colocalization analyses were located in transcriptionally active regions of the genome, we performed VSMC ATAC-seq (assay for transposase-accessible chromatin with sequencing) to delineate open chromatin regions, as well as DNA methylation assay and H3K27ac HiChIP-seq (histone H3 acetylation at lysine 27 high throughput chromosome conformation capture with chromatin immunoprecipitation with sequencing) since DNA demethylation and H3K27 acetylation upregulate gene expression. We interrogated the likely causal variants in regions of ATAC-seq peaks, low DNA methylation and H3K27ac, respectively. The analyses showed that a large proportion of the likely causal variants were located in regions of ATAC-seq open chromatin (Supplementary Table 9), low DNA-methylation (variants highlighted in yellow in Supplementary Table 10), and/or H3K27ac (Supplementary Table 11). The likely causal variants in relation to peaks of ATAC-seq, low DNA methylation and H3K27ac, as well as their associated eQTL genes and vascular diseases, are summarized in Supplementary Table 12. As an example, we found that the CAD associated variant rs1894401 in the *FES* gene resided in an ATAC-seq open chromatin region with lessened DNA-methylation (Supplementary Fig. 2&3) and with H3K27ac (Supplementary Fig. 4).” (page 4/paragraph 4).

3. The CRISPR screening identified important gene candidates, but the log fold change appears relatively small in this analysis.

As the reviewer rightly pointed out, the log fold change was relatively small from some of the genes. As the CRISPR screening is an in vitro experiment, it is unclear whether or not the effect sizes observed in this experimental setting are equivalent to the magnitudes of effects of these genes in vivo. However, as the pooled CRISPR screening can test multiple genes simultaneously under the same condition, it provides a ranking of the effects of the tested genes, based on a scoring method that has been widely used in many studies. We utilized this ranking to prioritize genes for validation experiments. We prioritized genes for validation using the following selection criteria: 1) among the top 10 genes ranked by score in the pooled CRISPR-Cas9-nuclease knockout screens; 2) considered to be a pleiotropic gene for more than one vascular disease in the cross-trait analysis; and 3) likely to be druggable in the druggability analysis (page 6/paragraph 3/line 6-9).

It is also unclear whether the y-axis represents p-value or adjusted p-value, and what the stringency of this analysis was.

The plots (Figure 5A&5B) show nominal p-values. We thank the reviewer for this comment and have modified the figure legends to clarify this. Adjusted p-values are included in Supplementary Table 24 & 25.

In Figure 5B, the blue and pink dots are marked as up in non-migrated cells, but further clarification is needed.

We apologize for the typo – the pink dots should have been marked as up in “migration cells” rather than “non-migrated cells”. We have corrected this typo.

4. The animal experiments indicate that downregulated expression of FES is associated with increased atherosclerotic lesions and hypertension. Are these results consistent with human data?

The results of the animal experiments are consistent with human data, both indicating that *FES* deficiency promotes atherosclerosis and hypertension. Specifically, the animal experiments showed that *Fes* knockout increased atherosclerotic lesion sizes and blood pressure (Figure 8), and the human data showed that *FES* gene deleterious variants are associated with increased risk of myocardial infarction and elevated blood pressure (Table 1, and page 7/paragraph 5).

To highlight this, we have revised the following text in the Discussion to read as follows:

“we demonstrated that knockout of *FES*, one of the pleiotropic genes, promotes atherosclerosis and raises blood pressure in mouse models, and additionally gathered evidence of *FES* gene deleterious variants significantly increasing the risk of both hypertension and CAD in humans among UK Biobank participants.” (page 9/paragraph 5/line 1-4).

Response to Reviewers' Comments

We thank the reviewers for their highly valuable comments and constructive advice. In the revised manuscript, the reviewers' points have been addressed as detailed below. Reviewers' comments are in bold font, followed by authors' response in regular font.

Reviewer 1

Colocalization strategy

For eCAVIAR and SMR/HEIDI analyses, can the author create a locus zoom window using the GWAS Tag SNP by building 200 KB (as a starting point) around significant SNPs based on LD, then adding 100 or 200 kb left or right until no SNP is in LD?

As advised, we have presented a locus zoom plot (Supplementary Figure 12). The plot shows that the GWAS tag SNP associated with CAD and all significant SNPs in LD with the GWAS tag SNP are located within a 100kb interval.

Fes knockout model and sex bias in in-vivo experiments:

Since in vitro FES gene analyses were conducted in VSMCs, unlike the in vivo mouse model with a whole-body knockout, it is difficult to determine the cell type source of the effect, making it hard to draw conclusive results. Therefore, it would be beneficial to have VSMC specific KO of Fes and study atherosclerosis.

We appreciate the reviewer's comment. However, the primary aim of our present study is not to test a cell-type specific effect on atherosclerosis and blood pressure, but to identify causal genes at the genomic loci that have been found to be associated with CAD, hypertension, stroke and/or AAA in GWAS. We have now also highlighted this in the Discussion as follows:

"... as *FES* is expressed in not only VSMCs but also some other cell types such as monocytes and endothelial cells, the total atherogenic contribution of *FES* is expected to derive from its effect on the various cell types, including VSMCs as suggested here and monocytes as indicated in our recent study²⁸. The relative contributions from the different cell types are yet to be established. Notwithstanding, the primary aim of our present study is not to test a cell-type specific effect on vascular diseases, but to identify causal genes at the genomic loci that have been found to be associated with CAD, hypertension, stroke and/or AAA in GWAS." (page 10/paragraph 3/line 12-18).

Additionally, combining data from both males and females with a small sample size, especially in females, for the atherosclerosis study, is a concern and makes it hard to identify if there is a sex-specific effect.

We thank the reviewer for the constructive comment. We have added a supplementary figure showing the data from males and females separately. (Supplementary Figure 10)

Reviewer 2

I appreciate that the authors addressed all my concerns.

We thank the reviewer for their time.

Reviewer 3

Report

Report focusing on the Fes knockout mouse experiments and the in-vitro validation data: The group has previously generated an eQTL catalogue for human umbilical cord smooth muscle cells

(SMC) and found co-localisation of SMC eQTLs with coronary artery disease (CAD)-associated risk loci. Here, they report overlap of hypertension, stroke and abdominal aortic aneurysm risk variants with SMC eQTLs and identify variant shared for multiple conditions. Co-localisation analysis and epigenetic profiling data is leveraged to identify candidate causal genes and associated mechanisms. The analysis suggests that some genes are linked to multiple conditions suggesting pleiotropic effects or common disease pathways for the conditions. Furthermore, database analysis highlights many of the identified genes as potential drug targets.

Pooled CRISPR screens of genes regulated by eQTLs colocalising with risk variants was done in human aortic smooth muscle cells to reveal impact on proliferation (enrichment at day 28 vs d 5) and migration (transwell migration). Validation experiments confirmed changes for 4 out of 5 genes selected based on suggested to have pleiotropic effects and top ranked in the CRISPR screen.

- **Significance (grey dots in Figure 5A) in the CRISPR screen appears to be evaluated without adjusting for multiple testing, which is problematic. The screen was done in triplicate, but it is not clear to me whether cells from different donors were used for the three screen replicates and how the replicates were considered for ranking of genes.**

We appreciate the reviewer's comment. FDR-adjusted P-values for the CRISPR knockout screen results are provided in Supplementary Tables 24 and 25. We would like to clarify that we performed these experiments on all the candidate genes simultaneously under the same condition, with the intention to prioritize some of these genes for further study. The prioritization was based on ranking of the tested genes. The ranking (further described below) was provided by the MAGeCK algorithm (Li *et al.*, Genome Biol 2014, PMID: 25476604) which we applied to analyze the pooled CRISPR knockout screen data from this study and has been widely used in many other studies for analyzing pooled CRISPR knockout screen data.

As described in the Methods (page 16/paragraph 1/line 2), the pooled CRISPR knockout screening experiments had 3 technical replicates. In these experiments, we included technical replicates (rather than biological replicates of cells from different donors), a strategy used in many other studies in the literature where pooled CRISPR screen experiments were performed using a single cell line in a study (e.g. Shalem *et al.*, Science 2014, PMID: 24336571; Wang *et al.*, Science 2014, PMID: 24336569; Konermann *et al.*, Nature 2015, PMID: 25494202; Joung *et al.*, Nat Protoc 2017, PMID: 28333914).

The MAGeCK algorithm integrates all the replicates in its statistical framework for gene ranking. The analysis begins by using the replicates to model the data with a Negative Binomial (NB) distribution and estimate the mean abundance, variance, and dispersion of sgRNA read counts. The algorithm uses the reads of non-targeting sgRNAs to normalize each replicate independently. The replicate-derived parameters are incorporated in the calculation of the log-fold change and P-value for each individual sgRNA. Thereafter, gene-level ranking is determined by the Robust Rank Aggregation (RRA) algorithm, which assesses the data across replicates by checking whether a gene's multiple sgRNAs are consistently and reproducibly ranked among the most enriched or depleted hits. In summary, replicates are not pooled but used independently to build a statistical reference for the initial sgRNA ranking, which later feed into RRA for final gene ranking.

- **For the CRISPR screen validation experiments, it is unclear how experimental replication is achieved; the legend to figure 5 states group sizes of 4 or 6, but methods section suggest that these are technical repeats.**

We apologize for unclearness of the text in the original figure legend and the confusion caused. As described in the Methods (page 17/paragraphs 1-4), they were technical replicates. We have now revised the figure legend to state that they are technical replicates.

For experiments shown in supplementary data, there are varying group sizes and timings (e.g. SMARCA evaluated at up to 96h). Ideally cells from different donors should be used for each experiment for datapoints to be considered independent.

For the prioritized genes, we used a different technique to validate the findings of the pooled CRISPR knockout screens. In the pooled CRISPR screens, each of the tested genes was knocked out which was mediated by CRISPR/Cas single-guide RNAs, and all of the studied genes were tested concurrently. In the validation experiments, each of the tested genes was knocked down by siRNA and each gene was tested individually. Thus, although the same source of cells was used in both the pooled CRISPR screens and validation experiments, a different technique was used in the validation experiment to assess the finding from the screen.

FES, which showed an impact on migration and genetic association with CAD, was selected for further analysis. The group has previously shown increased atherosclerosis in Fes knockout (KO) animals (PMID: 36321446) and this was here confirmed used an alternative measurement in the same experimental atherosclerosis model. Blood pressure measurements in non-atherogenic mice suggest that Fes-KO impacts both base line pressures and the response to acetylcholine. Furthermore, genetic variants resulting in truncated FES protein was associated with elevated blood pressure and likelihood of hypertension, angina and myocardial infarction in human.

- **The group size for blood pressure analysis is low, with only 4 animals in the KO group. This is a concern as the pressure is measured under anaesthesia (isoflurane, which can cause peripheral vasoconstriction), which is more variable. The low animal numbers also preclude comparison of effects in male and females.**

We appreciate the reviewer's comment. We have added the following text in the paragraph discussing limitations of our study:

"..., the experiments of *Fes* knockout mice were not intended and powered to test possible sex-specific effects of *Fes* on atherosclerosis and blood pressure". (page 10/paragraph 3/line 18-20).

- **The analysis of human data is not stratified by sex or analysed for confounding factors.**

The human genetic results used in our study were obtained from the AstraZeneca PheWAS Portal, which are based on the UK biobank cohort. These analyses are performed using established regression models that already adjust for key factors including sex (Backman *et al.*, Nature 2021, PMID: 34662886). The data we accessed are provided at the summary-statistic level, meaning that results are already pre-adjusted and further sex-stratified analyses cannot be carried out directly.

Molecular changes of siRNA-mediated FES knockdown in cultured SMCs are recorded by RNA-seq and proteomics analysis. This reveals changes in pathways and specific proteins, including MMP1.

- **The proteomics and phospho-proteomics only provide uncorrected p-values, with little or no, experimental validation.**

We appreciate the reviewer's comment and have added Benjamini-Hochberg adjusted P-values in Supplementary Tables 26 & 29.

- **It is not clear whether and how these changes are related to the in vivo phenotypes in mouse (or human traits), nor is the phenotype directly linked to an effect of FES on SMCs.**

We appreciate the reviewer's comment. While clarifying these points requires much further research, in this manuscript, we leverage these in vitro experimental results to help generate a mechanistic hypothesis.

Response to Reviewers' Comments

We thank the reviewers for their highly valuable comments and constructive advice. In the revised manuscript, the reviewers' points have been addressed as detailed below. Reviewers' comments are in bold font, followed by authors' response in regular font.

Reviewer 1

The authors addressed all my concerns.

We thank the reviewer for their time.

Reviewer 3

The revised manuscript have responded to most of the queries raised in my report, but the critique regarding group sizes for the in vivo analysis is not addressed (listed below).

We appreciate the reviewer's comment. In the revised manuscript, we have added the following text to explicitly acknowledge this limitation:

"Third, the *in vivo* experiments of *Fes* knockout were performed on small numbers of mice and were not powered to test possible sex-specific effects of *Fes* on atherosclerosis and blood pressure." (page 10, paragraph 3, lines 18-20)

The new information about replication and the appropriate statistical analysis raise some concerns regarding the robustness and reproducibility about the findings from the experimental analysis.

We appreciate the reviewer's comment. In the revised manuscript, we have added the following text:

"Fourth, the CRISPR knockout screens and follow-up experiments were carried out on VSMCs from a single donor and so could not account for potential inter-individual biological variations. Therefore, the findings from these experiments are exploratory and for hypothesis generation, rather than definitive." (page 10, paragraph 3, lines 20-23)

The revised manuscript state that the study does not aim to test whether the effects in VSMCs account for detected risk heritability, but to add a mechanistic hypothesis.

We understand that the reviewer considers this statement appropriate.

Additions to the revised manuscript

1. Appropriate statistical analysis (adjusted p-values) has been included for the crisp screen. Adjusted p-values suggest that most observed changes are unlikely to be due to actual biological differences. While FES scores 2nd in the migration screen, the adjusted p-value is 0.92.

We have modified the original text "The screens identified multiple genes that affected VSMC proliferation or migratory ability (Supplementary Table 24&25, and Fig. 5A&B)." which now reads "The results of these screens suggested that some of these genes might influence VSMC proliferation or migratory ability (Supplementary Table 24&25, and Fig. 5A&B)." (page 6, paragraph 3, lines 3-5)

2. The confusion regarding the nature of the replication has been resolved.

This reveals that all CRISPR screen replicates and the follow up experiments were done in

cells from a single donor. These two sets of experiments were done using different loss-of-function technologies. However, using a single genetic background means that the robustness of the observation is not tested.

We have added the following text to acknowledge this limitation:

“Fourth, the CRISPR knockout screens and follow-up experiments were carried out on VSMCs from a single donor and so could not account for potential inter-individual biological variations. Therefore, the findings from these experiments are exploratory and for hypothesis generation, rather than definitive.” (page 10, paragraph 3, lines 20-23)

3. The authors clarify that comparison between males and females is not intended. However, this does not address the concern regarding small group sizes.

We have added the following text to acknowledge this limitation:

“Third, the *in vivo* experiments of *Fes* knockout were performed on small numbers of mice and were not powered to test possible sex-specific effects of *Fes* on atherosclerosis and blood pressure.” (page 10, paragraph 3, lines 18-20)

4. Concern regarding confounding factors has been appropriately addressed.

5. Appropriate statistical analysis (adjusted p-values) has been included for the proteomics analysis. This reveals that most observed changes are unlikely to be due to actual biological differences, which does not justify the statement that “The phosphoproteomics analysis detected many differentially phosphorylated proteins (Supplementary Table 26)...”. However, the text has been revised to tone down the conclusions based on this data.

The original text “The phosphoproteomics analysis detected many differentially phosphorylated proteins (Supplementary Table 26) and a pathway analysis of these proteins suggested that they were enriched in several pathways including the epidermal growth factor (EGF)-, VEGF-, and CCKR (cholecystokinin A receptor)-signaling pathways (Fig. 6A). The RNA-seq and quantitative proteomics analyses indicated that *FES* knockdown led to changes in the production of multiple genes/proteins, including increased expression of the extracellular matrix protein-degrading enzymes matrix metalloproteinase-1 (MMP1) and MMP3 (Fig. 6B&C, Supplementary Table 27-29).” have been revised and now reads “The phosphoproteomics analysis identified a number of proteins whose phosphorylation status might be influenced by *FES* knockdown (Supplementary Table 26) and a pathway analysis of these proteins suggested that they were enriched in several pathways including the epidermal growth factor (EGF)-, VEGF-, and CCKR (cholecystokinin A receptor)-signaling pathways (Fig. 6A). The RNA-seq and quantitative proteomics analyses indicated that *FES* knockdown could potentially influence the production of a number of genes/proteins, including the extracellular matrix protein-degrading enzymes matrix metalloproteinase-1 (MMP1) and MMP3 (Fig. 6B&C, Supplementary Table 27-29).” (page 7, paragraph 1, lines 5-12)

Additionally, we have added the following text in the limitations paragraph in the Discussion:

“Therefore, the findings from these experiments are exploratory and for hypothesis generation, rather than definitive.” (page 10, paragraph 3, lines 22-23)

6. The discussion states that the manuscript does not intend to assess cell-type specificity.

We understand that the reviewer considers this statement appropriate.